# Precise gene regulation through transcriptional repression is essential for *Plasmodium berghei* asexual blood stage development

Tsubasa Nishi [1], Izumi Kaneko[2], Shiroh Iwanaga [1] & Masao Yuda [2] ✉

Malaria is caused by the proliferation of *Plasmodium* parasites in the vertebrate host blood stream through repeated cycles of asexual multiplication inside erythrocytes. During these cycles, parasites dynamically change their transcriptome at each developmental step to express genes exactly when required; however, the mechanisms regulating these transcriptomic changes remain unclear. In this study, we reveal that the AP2-family transcription factor PbAP2-TR is essential for the asexual blood stage development of the rodent malaria parasite *Plasmodium berghei*, as a transcriptional repressor. Conditional knockout of *pbap2-tr* causes developmental arrest at the trophozoite stage, *i.e.*, the cell growth phase of asexual blood stage development. The expression of PbAP2-TR target genes is upregulated in *pbap2-tr*-knockout parasites, and introduction of mutations into the binding motifs increases the promoter activity of the target genes. Time-course transcriptome analysis shows that PbAP2-TR is largely responsible for the transcriptional downregulation from early to late trophozoite development. Furthermore, our data demonstrate that PbAP2-TR recruits a putative chromatin remodeler, PbMORC, as a co-factor. These results indicate that transcriptional repression by PbAP2-TR-PbMORC complex contributes on precise control of transcription peak patterns during the asexual blood stage development.

*Plasmodium* is the causative agent of malaria, which is one of the most serious infectious diseases worldwide[1]. The parasite proliferates in host blood through repeated cycles of red blood cell (RBC) invasion by a specific invasive stage called the merozoite and asexual replication inside host RBCs[2–4]. Intraerythrocytic asexual development can be divided into three stages: ring, trophozoite, and schizont[5]. The ring stage, which is the first developmental stage formed after merozoite invasion, is characterized by a biconcave disc shape[6]. At the ring stage, the expression of exported proteins is upregulated to modify the host cell into a parasite-favorable environment[7]. The next developmental stage, trophozoite, grows rapidly with an elevated uptake of host-derived nutrients, including hemoglobin, which is a source of amino acids[8,9]. For rapid cell growth, gene transcription becomes considerably more active during the trophozoite stage than during the ring stage[10]. The parasite then develops into the schizont stage, which undergoes a unique mode of cell division called schizogony[11,12]. In schizonts, several rounds of asynchronous nuclear division occur while merozoite components, such as the rhoptry and inner membrane complex (IMC), are generated[13]. Finally, multiple progeny merozoites are formed through segmentation simultaneously with a final round of nuclear division.

[1]Research Institute for Microbial Diseases, Osaka University, Suita, Japan. [2]Department of Medicine, Mie University, Tsu, Japan.
✉e-mail: m-yuda@med.mie-u.ac.jp

As the asexual blood stage development proceeds, the parasite dynamically alters gene expression. Time-course transcriptome analyses using microarray and steady-state RNA sequencing (RNA-seq) have shown that the transcript levels of genes that have a common functional role show similar peak patterns in the course of the intraerythrocytic developmental cycle (IDC)[14–16]. Such temporal changes in gene transcription largely appear as the cascade of transcriptional activation to express genes just when they are required, *i.e.*, "just-in-time" transcription. Concordant with the transcription peak patterns, nucleosome-free or nucleosome-depleted regions, which indicate active transcription sites, are also periodically established[17]. Furthermore, a recent study indicated the essentiality of chromatin remodelers for the "just-in-time" regulation of gene expression during the asexual blood stage development[18]. These suggest that regulation at the transcriptional level is important for IDC progression; thus, investigating the mechanisms that underlie this periodic transcriptional regulation is very important to obtain a deep understanding of parasite asexual proliferation in the host blood.

*Plasmodium* spp. have a limited number of sequence-specific transcription factors encoded in their genome. These transcription factors are mainly AP2 family proteins, which are homologues of the plant APETALA2/Ethylene Response Factor (AP2/ERF) and contain one to three DNA-binding domains called AP2 domains[19]. In previous studies, several stage-specific transcription factors have been explored, and extensive progress has been particularly made regarding the development of invasive and sexual stages[20]. At these stages, transcriptional activation of stage-specific genes is controlled by a few sequence-specific transcription factors and their corresponding binding motifs. For the asexual blood stage development, two transcriptional activators have been reported, PfAP2-I and PbSIP2[21,22]. PfAP2-I is expressed from trophozoites to early schizonts, and targets invasion-related genes during the schizont stage. PbSIP2 functions as a master regulator essential for merozoite formation, comprehensively activating genes related to merozoite formation, such as genes related to the rhoptry, microneme, and IMC. Meanwhile, transcription factors that control transcriptomic changes during the other stages of the IDC, *i.e.* ring and trophozoite stages, have not been identified, and the mechanism regulating the transcription peak patterns for each gene remains largely unknown.

To advance our knowledge of transcriptional regulation during the IDC, we focused on AP2 transcription factor genes that could not be knocked out during the IDC of *P. berghei*. Here, we demonstrate that the AP2 family transcription factor, PbAP2-TR (Trophozoite Repressor, PBANKA_0909600), functions as a transcriptional repressor essential for trophozoite development. Our data reveal that PbAP2-TR represses the transcription of its target genes during trophozoite development and controls their transcriptional profiles. Furthermore, we show that PbAP2-TR recruits a putative chromatin remodeler, PbMORC, as a possible co-factor for the transcriptional repression of target genes.

## Results

### PbAP2-TR is an AP2 transcription factor expressed in trophozoites

PbAP2-TR is an AP2-family transcription factor that possesses three conserved AP2 domains (Fig S1A). The first (position 182–235) and second (position 279–325) AP2 domains are located in tandem near the N-terminus, and the third (position 1474–1522) domain is located towards the C-terminus. When the amino acid (AA) sequences of the tandem AP2 domains were aligned among PbAP2-TR orthologs in *Plasmodium* species, both the AP2 domains and the linker between them were found to be highly conserved (96% and 87% for the first and second AP2 domains, respectively; 81% for the linker) (Fig S1B). For the third AP2 domain of PbAP2-TR, the AA sequences were completely conserved within *Plasmodium* (Fig S1C). In addition to these AP2 domains, PbAP2-TR contains a conserved AP2-coincident domain mostly at the C-terminus (ACDC) and a putative nuclear localization signal (NLS) at position 1139–1151 (Fig S1A).

We first attempted to generate *pbap2-tr*-knockout parasites using conventional homologous recombination[23] but failed to obtain mutant parasites. This was consistent with previous knockout screening studies on rodent malaria parasites[24–26]. These results indicate that *pbap2-tr* is essential for asexual blood stage development. To evaluate at which cell-stage PbAP2-TR functions during intraerythrocytic development, we assessed the PbAP2-TR expression pattern. We generated a parasite line expressing GFP-fused PbAP2-TR (PbAP2-TR::GFP, Fig S2) using the Cas9-expressing parasite, PbCas9[27], and performed a time-course fluorescent analysis. The cell cycle was synchronized to mature schizonts by incubating PbAP2-TR::GFP-infected blood in culture for 16 h, and the cultured schizonts were then intravenously injected into mice. A nuclear-localized GFP signal was first observed in early trophozoites at 8 h post-injection (hpi) (Fig. 1A). The signal continued to be observed for the late trophozoites in the peripheral blood (Fig. 1A). As later asexual blood stages are rarely observed in peripheral blood owing to sequestration, we further performed fluorescence analysis on cultured PbAP2-TR::GFP. In the culture, mononuclear parasites showed nuclear-localized GFP signals, as observed in peripheral blood; however, the signal was considerably weaker in schizonts with two nuclei. Furthermore, later schizonts showed no GFP signal (Fig. 1A). These results indicate that PbAP2-TR is a trophozoite-specific transcription factor.

### Disruption of *pbap2-tr* results in developmental arrest at the trophozoite stage

As gene disruption of *pbap2-tr* was unsuccessful, we performed a conditional knockout of *pbap2-tr* to assess the role of PbAP2-TR. We employed a previously reported dimerizable Cre (DiCre) system[28,29] using a parasite line that constitutively expressed Cas9 and DiCre (PbCas9^DiCre)[22]. Two loxP sequences were each introduced to the 5′ and 3′ sides of *pbap2-tr*, arranged in the same direction (*pbap2-tr*-DiCre, Fig S3A), for removing the *pbap2-tr* locus in a rapamycin-dependent manner (Fig. 1B). For the conditional knockout, whole blood was harvested from mice infected with *pbap2-tr*-DiCre and split into two cultures (Fig. 1C). At the beginning of the culture, rapamycin was added to one culture (final concentration, 30 nM) (*pbap2-tr*-DiCre^Rapa+), and dimethyl sulfoxide (DMSO) was added to the other (*pbap2-tr*-DiCre^Rapa-) as a control (Fig. 1C). After 16 h of culture, genotyping PCR confirmed that the *pbap2-tr* locus was almost completely excised from the genome of *pbap2-tr*-DiCre^Rapa+, with only a subtle signal for the wild-type (WT) *pbap2-tr* locus (Fig. 1D). To assess the effect of *pbap2-tr*-disruption, mature schizonts of *pbap2-tr*-DiCre^Rapa+ and *pbap2-tr*-DiCre^Rapa- were inoculated into mice (Fig. 1C). The parasites were then cultured at 10 hpi, and parasite development was assessed by Giemsa-staining every four hours (Fig. 1C). In the cultures, *pbap2-tr*-DiCre^Rapa- and *pbap2-tr*-DiCre^Rapa+ showed similar ring-to-trophozoite ratios until 18 hpi (Fig. 1E). In addition, the average size of trophozoites at 18 hpi was comparable between *pbap2-tr*-DiCre^Rapa- and *pbap2-tr*-DiCre^Rapa+ with a diameter of approximately 4.4 μm (Fig. 1F). However, at 22 hpi, the schizont ratio reached 20% in *pbap2-tr*-DiCre^Rapa-, whereas most of the *pbap2-tr*-DiCre^Rapa+ parasites remained as mononuclear cells (schizont ratio: 0.6%) (Fig. 1E). Furthermore, the ratio of schizonts in *pbap2-tr*-DiCre^Rapa+ was still less than 3% at 26 hpi, whereas that in *pbap2-tr*-DiCre^Rapa- was higher than 60% (Fig. 1E). This suggests that PbAP2-TR functions during trophozoite development before the beginning of nuclear division.

To further confirm the essentiality of *pbap2-tr* in asexual blood stage development, we assessed the parasite growth of *pbap2-tr*-DiCre in vivo. After inducing rapamycin-dependent knockout and inoculating the schizonts of *pbap2-tr*-DiCre^Rapa- and *pbap2-tr*-DiCre^Rapa+ into mice, parasitemia was determined through daily Giemsa staining. Parasite infectivity immediately after inoculation was comparable

between *pbap2-tr*-DiCre[Rapa-] and *pbap2-tr*-DiCre[Rapa+] as their parasitemia was not significantly different on day 0 (6 hpi), suggesting that *pbap2-tr*-disruption did not affect the ability of schizonts/merozoites to infect host RBCs (Fig. 1G). On day 1, the parasitemia of *pbap2-tr*-

DiCre[Rapa+] decreased from that on day 0 and was significantly lower than that of *pbap2-tr*-DiCre[Rapa-] (Fig. 1G). In the following days, parasitemia of *pbap2-tr*-DiCre[Rapa+] increased with a growth rate comparable to that of *pbap2-tr*-DiCre[Rapa-] (Fig. 1G). However, genotyping on day 3

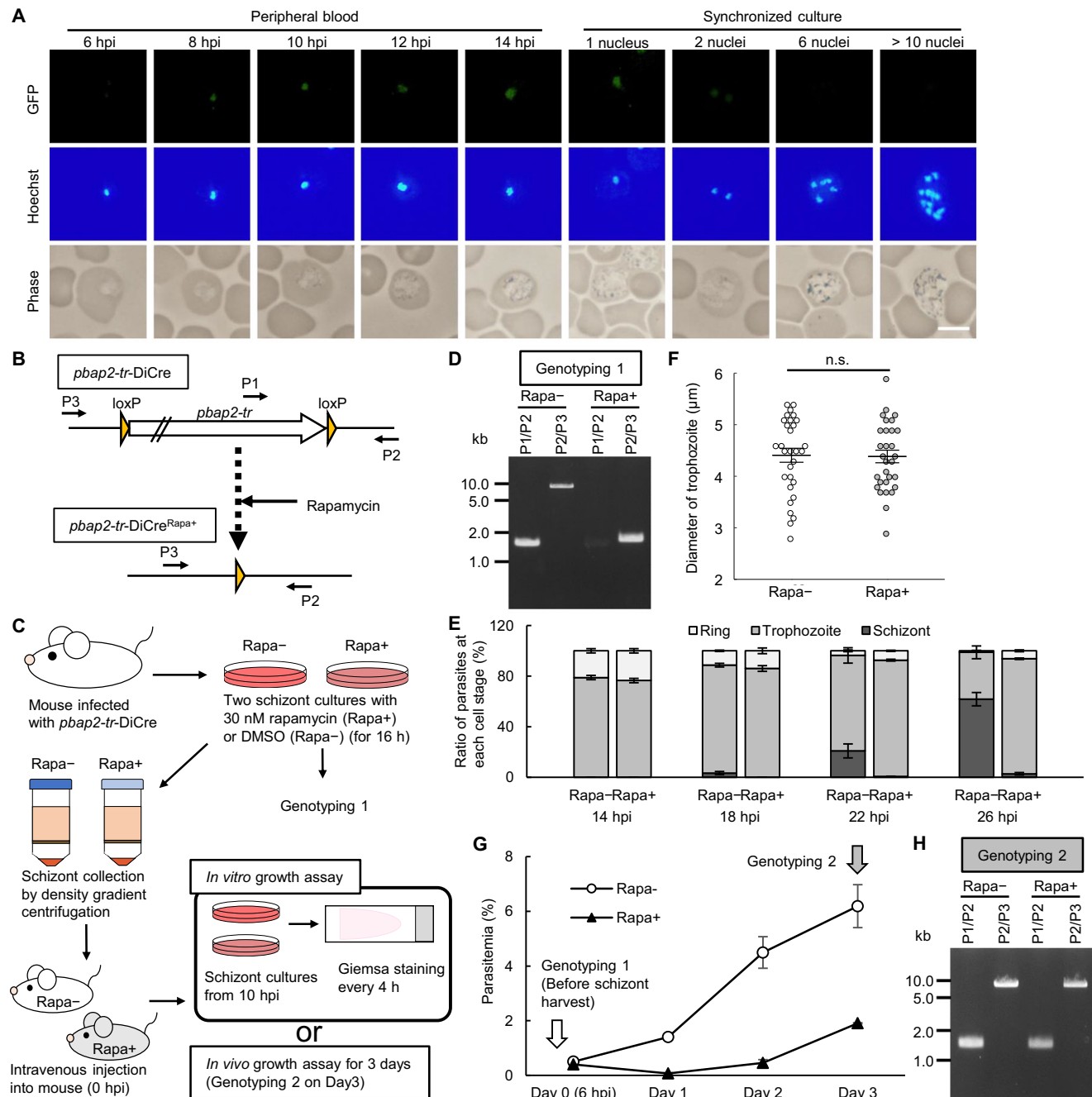

**Fig. 1 | Conditional knockout of *pbap2-tr* using the DiCre system. A** Expression pattern of PbAP2-TR::GFP during asexual blood stage development. Parasites were observed in the peripheral blood from 6–14 hours post-injection (hpi) and in cultures for assessing schizonts. Nuclei were stained with Hoechst 33342. Scale bar = 5 μm. **B** Schematic of the *pbap2-tr* locus in the *pbap2-tr*-DiCre and rapamycin-induced DiCre recombination. Primer positions (P1–P3) used for genotyping PCR are indicated. **C** Outline of the conditional *pbap2-tr* knockout. Blood from mice infected with *pbap2-tr*-DiCre was split into rapamycin-treated (Rapa + ) and control (Rapa − ) cultures. After 16 h, schizonts were purified and injected intravenously into mice (0 hpi). For assessing in vitro, infected blood harvested at 10 hpi was cultured again, and then parasite development was assessed by Giemsa-staining every 4 hours. For assessing in vivo, parasitemia was assessed daily. **D** Representative genotyping PCR analysis for Rapa− and Rapa+ at 16 h of culture

from three biologically independent experiments (Genotyping 1). **E** Ratio of ring, trophozoite, and schizont stages in Rapa− and Rapa+ cultures over time. Error bars indicate SEM from three biologically independent experiments. **F** Trophozoite size in Rapa− and Rapa+ at 18 hpi. Longest diameter was measured on Giemsa-stained smears (n = 30: ten trophozoites were assessed for each sample in three biologically independent experiments, and the data from Rapa− and Rapa+ were each pooled.). Lines indicate mean values and SEM. (n.s.: not significant, two-tailed Student's t-test, p-value > 0.05.) **G** Parasitemia of Rapa− and Rapa+ during in vivo growth. Error bars indicate SEM from three biologically independent experiments.
**H** Representative genotyping PCR analysis for Rapa− and Rapa+ on day 3 from three biologically independent experiments (Genotyping 2). Source data are provided as a Source Data file.

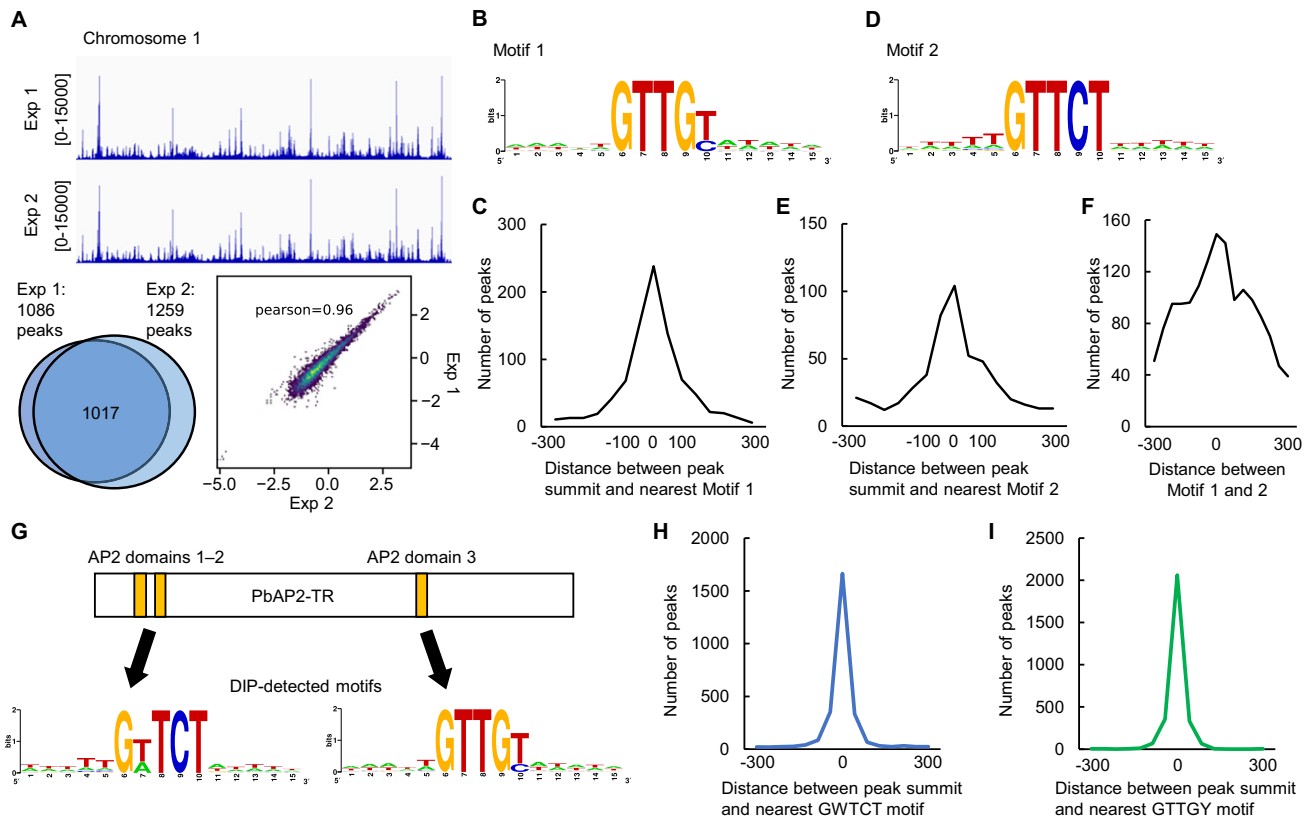

**Fig. 2 | DNA binding properties of PbAP2-TR assessed by chromatin immuno-precipitation followed by high-throughput sequencing (ChIP-seq) and DNA immunoprecipitation followed by high-throughput sequencing (DIP-seq) analyses. A** Integrative Genomics Viewer images for the PbAP2-TR ChIP-seq experiments 1 and 2 on chromosome 1. Histograms show the raw read coverage of ChIP data normalized by library size (bin size = 10 bp). Scales are indicated in square brackets. A Venn diagram at the left bottom shows the number of overlapping peaks between experiments 1 and 2. A scatter plot at the right bottom shows comparison of log2(ChIP/input) in replicate at each 1-kb bin. **B** The motif most enriched within 50 bp from the summits of the PbAP2-TR ChIP-seq peaks (Motif 1).

The logo was generated using WebLogo (https://weblogo.berkeley.edu/logo.cgi). **C** Distance between the peak summits and the nearest Motif 1. **D** The second most enriched motif for PbAP2-TR ChIP-seq peaks (Motif 2). **E** Distance between the peak summits and the nearest Motif 2. (**F**) Distance between Motif 1 and 2 in the peak region. **G** Motifs enriched in the DIP-seq peak regions for AP2 domains 1–2 (left) and domain 3 (right). The schematic illustration of PbAP2-TR is shown at the top. The logos were generated using WebLogo. **H** Distance between DIP-seq peak summits for AP2 domains 1–2 and the nearest GWTCT (W = A or T) motifs. **I** Distance between the DIP-seq peak summits for AP2 domain 3 and the nearest GTTGY motifs.

did not detect *pbap2-tr*-disrupted parasites but only parasites with the WT *pbap2-tr* locus (Fig. 1H). This result confirms that *pbap2-tr* plays an essential role in parasite development during the IDC.

Previously, Shang *et al.* reported that in *P. falciparum*, disruption of the *pbap2-tr* ortholog, named *pfap2-g5*, resulted in upregulation of PfAP2-G and increased the rate of differentiation into gametocytes, consequently causing a significant reduction in the parasite asexual growth rate[30]. Thus, they concluded that PfAP2-G5 is important for suppressing commitment to the gametocyte fate. To evaluate whether PbAP2-TR plays a similar role, we assessed the number of gametocytes in *pbap2-tr*-DiCre[Rapa-] and *pbap2-tr*-DiCre[Rapa+] on day 1 (24 hpi) of the above in vivo growth assays (because gametocyte maturation takes approximately 24 h in *P. berghei*) through Giemsa-staining. Gameto-cytemia (gametocyte/RBC) was comparable between *pbap2-tr*-DiCre[Rapa-] and *pbap2-tr*-DiCre[Rapa+], demonstrating that the rate of commitment to gametocyte fate was not significantly affected by whether *pbap2-tr* was disrupted (Fig S3B). Next, to further confirm that the loss of asexual proliferation in *pbap2-tr*-DiCre[Rapa+] was not due to the parasite commitment to gametocyte fate, we disrupted *pbap2-g* in *pbap2-tr*-DiCre using the CRISPR/Cas9 system [*pbap2-tr*-DiCre[pbap2-g(-)], Fig S3C]. DiCre-mediated disruption of *pbap2-tr* was induced in cultures (*pbap2-tr*-DiCre[pbap2-g(-)_Rapa+] and *pbap2-tr*-DiCre[ap2-g(-)_Rapa-]) (Genotyping 1, Fig S3D left), and after inoculating *pbap2-tr*-DiCre[pbap2-g(-)_Rapa-] and *pbap2-tr*-DiCre[ap2-g(-)_Rapa+] into mice, parasitemia was assessed daily.

The result was similar to that of the phenotype analysis for *pbap2-tr*-DiCre; in *pbap2-tr*-DiCre[ap2-g(-)_Rapa+], parasitemia decreased from day 0 to day 1 and later increased (Fig S3E), but the day-3 parasites possessed the WT *pbap2-tr* locus (Genotyping 2, Fig S3D right). Collectively, these results suggest that PbAP2-TR is indeed essential for asexual blood stage development and does not play any evident role in the sexual commitment unlike PfAP2-G5.

## PbAP2-TR recognizes two DNA motifs using different AP2 domains

To investigate the target genes regulated by PbAP2-TR, we performed chromatin immunoprecipitation followed by high-throughput sequencing (ChIP-seq) of PbAP2-TR at the trophozoite stage. In *P. falciparum*, PfAP2-G5 also plays a role in the early gametocyte development. Thus, to specifically assess the role of PbAP2-TR in asexual development, we disrupted *pbap2-g* in PbAP2-TR::GFP using the CRISPR/Cas9 system [PbAP2-TR::GFP[pbap2-g(-)], Fig S4A]. ChIP-seq analyses were performed in duplicate using PbAP2-TR::GFP[pbap2-g(-)] at 12 hpi. Experiments 1 and 2 detected 1086 and 1259 peaks, respectively (by macs2 with the thresholds of fold enrichment > 3.0 and *q*-value < 0.01, using sequence data of input DNA as a control), and 1017 peaks overlapped (93% of the peaks in experiment 1) (Fig. 2A, Supplementary Data 1). Genome-wide comparison of ChIP/input enrichment between the replicates resulted in Pearson correlation coefficient of 0.96, and

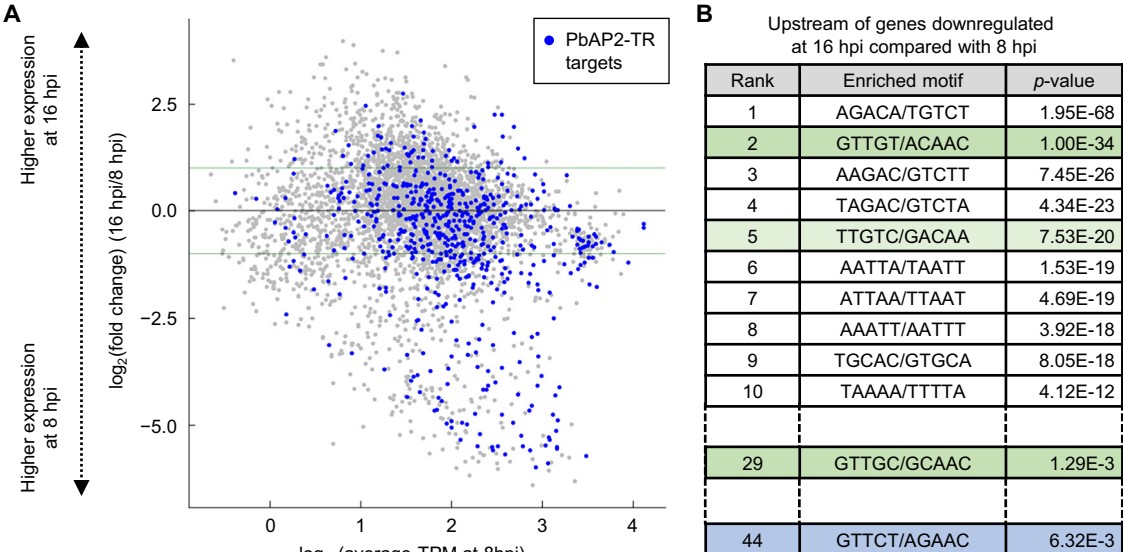

**Fig. 3 | Transcriptional changes in PbAP2-TR targets during the PbAP2-TR expression period. A** MA plot showing the differential expression of genes from 8 to 16 hpi. Blue dots represent the PbAP2-TR targets. The two horizontal lines indicate the $\log_2$(fold change) of 1 and −1. **B** DNA motifs enriched in the upstream regions (200–1000 bp from the start codons) of genes downregulated from 8 to 16 hpi. $p$-values were calculated using Fisher's exact test. Motif 1 (GTTGY) and Motif 2 (GTTCT) are indicated in green and blue, respectively. Motifs partially overlapped with Motif 1 are indicated in light green.

IDR1D analysis revealed rank of peaks in each replicate to be largely consistent, showing high reproducibility of the data (Fig. 2A, S4B, and S4C). From the common peaks, we first explored the putative binding motifs of PbAP2-TR by DNA motif enrichment analysis using Fisher's exact test. The analysis revealed GTTGTA as the most enriched motif ($p$-value = $5.2 \times 10^{-72}$) and several motifs containing GTTGT in the 20 most enriched motifs (Supplementary Data 1). Furthermore, the analysis detected the enrichment of motifs containing GTTGC. Thus, the most enriched motif was considered as GTTGY (Fig. 2B). Among the 1017 ChIP-seq peaks, 872 peaks contained the GTTGY motif within 300 bp of the summit, and most were found within 100 bp (Fig. 2C). In addition to GTTGY, motifs containing GTTCT were highly enriched (Fig. 2D). This motif was found in 514 peaks, mostly close to the summit (Fig. 2E). Similar motifs were also enriched in the ChIP-seq analysis of PfAP2-G5[30]. Hereafter, the GTTGY and GTTCT motifs are referred to as Motif 1 and 2, respectively. In total, 90% of the ChIP-seq peaks (934 peaks) contained at least either Motif 1 or 2, and nearly half of the peaks (452 peaks) contained both of the motifs. Among the peaks containing both Motifs 1 and 2, the distance between these two motifs was not constant, *i.e.* these motifs are not arranged in a specific positional relationship (Fig. 2F).

To further evaluate the DNA-binding properties of PbAP2-TR, we performed DNA immunoprecipitation followed by high-throughput sequencing (DIP-seq) analysis using recombinant PbAP2-TR AP2 domains fused with maltose-binding protein (MBP) and fragmented *P. berghei* genomic DNA. First, DIP-seq was performed on two N-terminal tandem AP2 domains. Because these two AP2 domains likely function together owing to their proximity and the conserved amino acids between them (Fig S1B), we generated a recombinant protein containing both AP2 domains. In this analysis, 3038 peaks were detected throughout the genome (Supplementary Data 2). Motif enrichment analysis of these DIP peaks detected GTTCT and GATCT as the two most enriched motifs (Fig. 2G and Supplementary Data 2), which was consistent with the previously identified binding motif for the first AP2 domain of PfAP2-G5[31]. Among the 3038 DIP peaks, 2502 peaks (82.3%) contained GWTCT (W = A or T) within 100 bp of their summit (Fig. 2H), suggesting that GWTCT, which is analogous to Motif 2, is the binding motif of the N-terminal tandem AP2 domains of PbAP2-TR. Next, DIP-

seq was performed on the third AP2 domain of PbAP2-TR. In this analysis, 2980 peaks were detected (Supplementary Data 2), and the most enriched motif in these peak regions was GTTGT (Supplementary Data 2). The following enriched motifs were one-base-shifted motifs of GTTGT (TGTTG, AGTTG, and TTGTA) and GTTGC (Supplementary Data 2). Thus, the enriched motifs were related to Motif 1 (GTTGY) (Fig. 2G). Motif 1 was contained within 100 bp from the summit of 2888 peaks (97%) (Fig. 2I). Collectively, these results suggest that PbAP2-TR binds to Motif 1 and 2 each by different AP2 domains.

## PbAP2-TR targets are downregulated during early to late trophozoite development

Based on ChIP-seq data, we identified 619 target genes of PbAP2-TR, which were defined as genes with a ChIP-seq peak within 1200 bp of the start codon (Supplementary Data 3). Among these targets, only 88 genes overlapped with those of PfAP2-G5 at the trophozoite stage (517 genes reported by Shang *et al.*) (Supplementary Data 3). To examine the role of PbAP2-TR in transcriptional regulation, we investigated the changes in the transcript levels of these target genes from the beginning (8 hpi) to the end (16 hpi) of PbAP2-TR expression using high-throughput RNA sequencing (RNA-seq). To obtain asexual transcriptomic data, RNA-seq analyses were conducted using gametocyte-less mutant parasites, *i.e.*, the *ap2-g*-knockout line, in biological triplicate (quality check results were shown in Fig S5A–C). Among the genes significantly downregulated at 16 hpi compared with those at 8 hpi (799 genes in total, $\log_2$(fold change) < −1, $p$-value adjusted for multiple testing with the Benjamini-Hochberg procedure ($p$-value^adj < 0.05), 180 genes were PbAP2-TR targets whereas only 55 targets were included in the significantly upregulated genes (702 genes in total, $\log_2$(fold change) > 1, $p$-value^adj < 0.05) (Fig. 3A and Supplementary Data 4). Overall, the average $\log_2$(fold change) value of the PbAP2-TR target genes was significantly lower than that of the other genes ($p$-value = $2.3 \times 10^{-26}$ by two-tailed Student's t-test). In addition, in the upstream region (200–1000 bp from the start codon) of the downregulated genes, GTTGT (Motif 1) was the second most significantly enriched with a $p$-value of $1.0 \times 10^{-34}$ by Fisher's exact test (Fig. 3B). The other binding motifs, GTTGC (Motif 1) and GTTCT (Motif 2), were also enriched with $p$-values of $1.3 \times 10^{-4}$ and $6.3 \times 10^{-3}$, respectively (Fig. 3B).

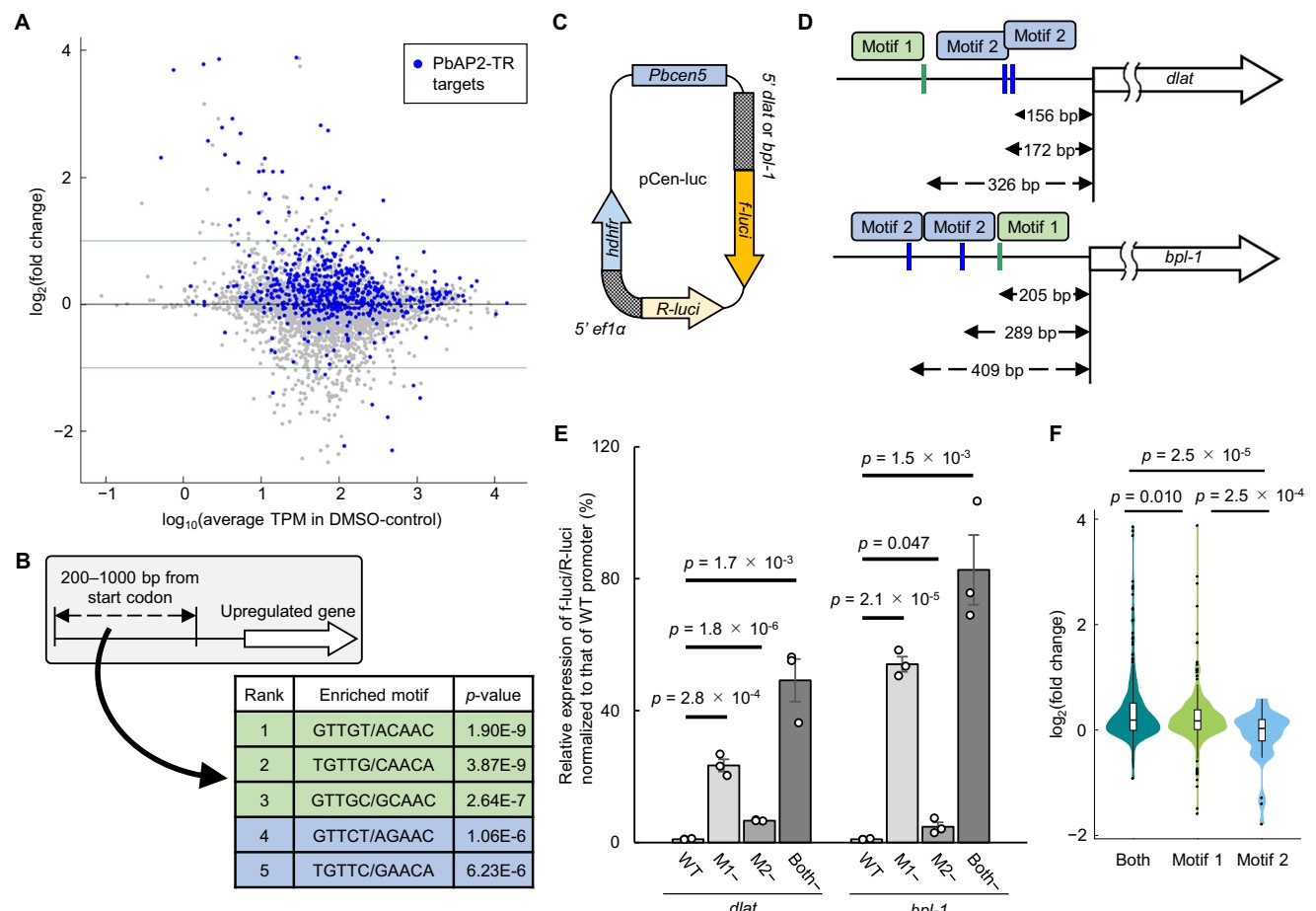

**Fig. 4 | Function of PbAP2-TR as a transcriptional repressor. A** MA plot showing differential gene expression between *pbap2-tr*-DiCre^Rapa−^ and *pbap2-tr*-DiCre^Rapa+^ at 12 hours post-injection. Blue dots represent PbAP2-TR targets. Horizontal lines indicate log₂(fold change) of 1 and −1. **B** Enrichment of DNA motifs in the 200–1000 bp upstream regions of genes upregulated in *pbap2-tr*-DiCre^Rapa+^. Motifs related to Motif 1 (GTTGY) and Motif 2 (GTTCT) are indicated in green and blue, respectively. **C** Schematic illustration of the pCen-luc plasmid (*f-luci*, firefly luciferase gene, R-luci, Renilla luciferase gene; hdhfr, human dihydrofolate reductase gene; *5′ ef1α*, bi-directional promoter of *P. berghei* elongation factor 1-alpha genes; *Pbcen5*, the centromere of *P. berghei* chromosome 5). **D** Schematic illustrations of the *dlat* and *bpl-1* loci. Locations of Motif 1 and 2 are indicated with distances from the start codon. **E** Dual-luciferase reporter assays for promoter activities of *dlat* and

*bpl-1*. Relative luminescence (f-luci/R-luci) was normalized to the corresponding wild-type reporter (WT). M1− and M2− denote reporters with mutations in Motif 1 and Motif 2, respectively. Both− is a reporter with mutations in both motifs. Dots represent f-luci/R-luci values for each replicate. Error bars indicate the SEM from three biologically independent experiments. *p*-values were calculated using a two-tailed Student's t-test (n.s.: not significant, *p*-value > 0.05). **F** Violin plot showing distribution of log₂(fold change) values for three groups of PbAP2-TR targets: those with ChIP-seq peaks containing both Motif 1 and 2 (n = 283), only Motif 1 (n = 261), and only Motif 2 (n = 40). *p*-values were calculated using a two-tailed Student's t-test. The corresponding box plots show the median (center line), 25th/75th percentiles (box), and minimum/maximum values (whiskers). The dots indicate outliers. Source data are provided as a Source Data file.

These results suggest that PbAP2-TR is the major transcriptional repressor responsible for gene downregulation during early to late trophozoite development.

## PbAP2-TR functions as a transcriptional repressor

To verify that PbAP2-TR is a transcriptional repressor, we performed a differential expression analysis between *pbap2-tr*-DiCre^Rapa−^ and *pbap2-tr*-DiCre^Rapa+^ at 12 hpi (quality check results were shown in Fig S6A–C). The analysis revealed that 70 and 111 genes were significantly upregulated and downregulated, respectively, in *pbap2-tr*-DiCre^Rapa+^ compared with those in *pbap2-tr*-DiCre^Rapa−^ (Fig. 4A, S6D, and Supplementary Data 5). Of note, *pbap2-g* was not differentially expressed in *pbap2-tr*-DiCre^Rapa+^ (log2(fold change) = 0.075 and *p*-value^adj^ = 0.79) consistent with the comparable sexual commitment rate between *pbap2-tr*-DiCre^Rapa−^ and *pbap2-tr*-DiCre^Rapa+^ (Fig S3B). Among the upregulated genes, 47 PbAP2-TR target genes were detected, showing significant enrichment with a *p*-value of $4.2 \times 10^{-21}$ by Fisher's exact test (Fig. 4A). In contrast, the downregulated genes

included only nine target genes (Fig. 4A). Furthermore, the average log₂(fold change) value for the PbAP2-TR target genes was significantly higher than that for the other genes with a *p*-value of $2.6 \times 10^{-67}$ by two-tailed Student's t-test (Fig S1D). Motif enrichment analysis showed enrichment of both Motif 1 and 2 in the upstream sequences (200–1000 bp from the start codon) of the upregulated genes compared with those of the other genes (Fig. 4B). These results confirm that PbAP2-TR functions as a transcriptional repressor during trophozoite development.

## Binding motifs of PbAP2-TR function as *cis*-regulatory repressing elements

To evaluate the functions of Motif 1 and 2 as *cis*-regulatory elements, we performed dual luciferase assays using a centromere plasmid containing firefly and *Renilla* luciferase genes (pCen-luc, Fig. 4C). Expression of the *Renilla* luciferase (R-luci) gene in pCen-luc was controlled by the *P. berghei* elongation factor 1-alpha promoter as an internal control, and the target gene promoters were inserted

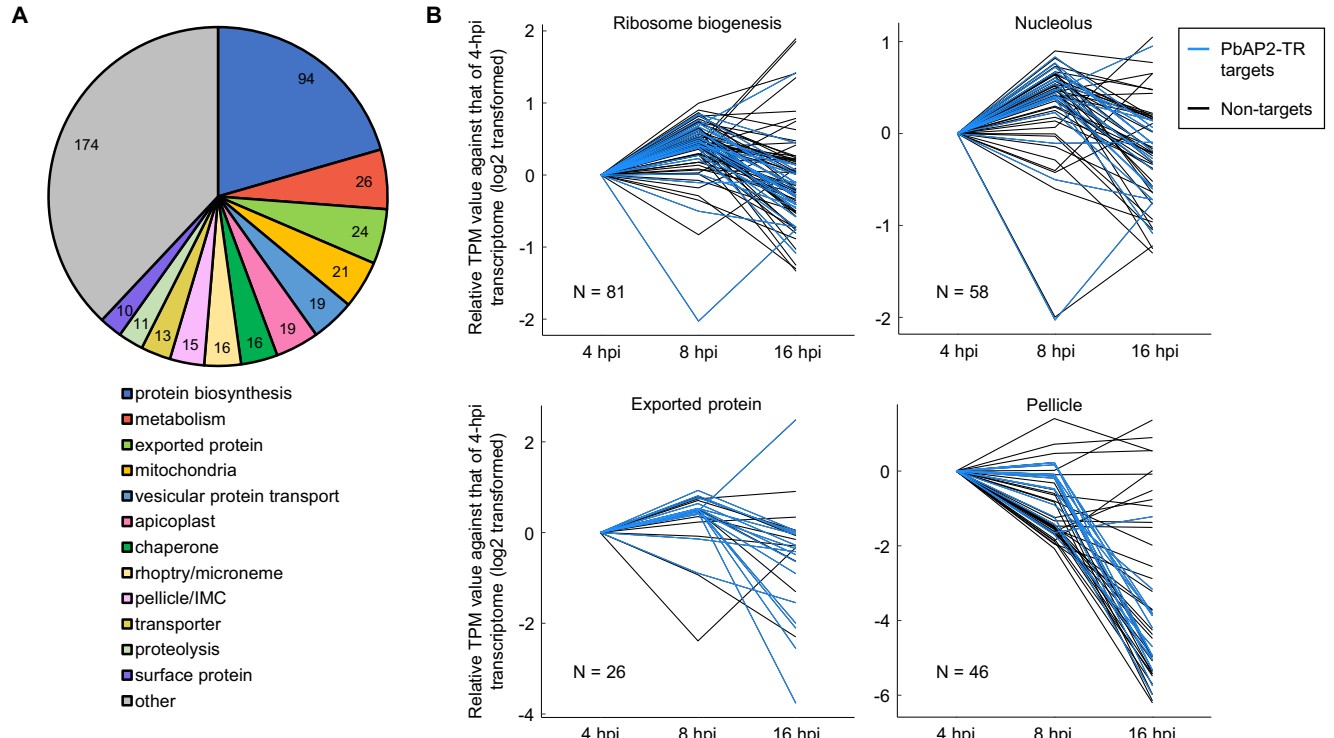

**Fig. 5 | Functional profiling of PbAP2-TR targets and the expression patterns of target-enriched gene sets. A** Classification of PbAP2-TR target genes into 13 characteristic groups. Genes annotated on the PlasmoDB (https://plasmodb.org/plasmo/app) (458 genes) were classified. **B** Expression patterns of gene sets enriched in the PbAP2-TR targets. Transcriptomic analyses were performed in biological triplicate at each time-point. Transcripts per million (TPM) values at 4, 8, and 16 hpi are shown as relative values normalized to those of 4 hpi. N indicates the number of genes in each set (genes with TPM less than 30 at all time-points were excluded). The PbAP2-TR targets are highlighted in blue.

upstream of the firefly luciferase (f-luci) gene (Fig. 4C). We selected two PbAP2-TR target genes, dihydrolipoamide acyltransferase (*dlat*, PBANKA_0505000) and biotin--protein ligase 1 (*bpl-1*, PBANKA_0511000), for the reporter assays, as these were significantly upregulated upon *pbap2-tr*-disruption (Supplementary Data 5). The upstream regions of *dlat* and *bpl-1* both contained one of Motif 1 and two of Motif 2 (Fig. 4D), and we introduced mutations to alter either or both of these motifs from GTTGY and GTTCT to tTTGY and tTTCT, respectively. These pCen-luc plasmids were then introduced into the wild-type *P. berghei* ANKA strain, and dual luciferase assays were performed at 14 hpi in biological triplicate.

In the assay for the *dlat* promoter, the relative expression of f-luci against R-luci was significantly increased by 23- and 7-fold through mutations into Motif 1 and 2, respectively (*p*-value < 0.01, two-tailed Student's t-test) (Fig. 4E). Furthermore, f-luci/R-luci expression was the highest with mutations into both Motifs 1 and 2 (49-fold higher than that without mutations) (Fig. 4E). In the assay for *bpl-1*, a similar result was obtained; promoter activity significantly increased with mutations into either Motif 1 or 2 (by 54- and 5-fold, respectively), and mutations into both motifs induced the highest f-luci/R-luci expression level (83-fold higher than that without mutations) (Fig. 4E). Thus, for both reporters, Motifs 1 and 2 both functioned as *cis*-regulatory elements for transcriptional repression, and Motif 1 contributed more to transcriptional repression than Motif 2.

Next, we evaluated whether the functional difference between Motifs 1 and 2 was also observed in the differential expression analysis between *pbap2-tr*-DiCre^Rapa+^ and *pbap2-tr*-DiCre^Rapa-^. We divided the target genes into three groups: targets of a peak with both motifs, only Motif 1, and only Motif 2, within 300 bp of the summit (Supplementary Data 5). The average log$_2$(fold change) value of the "both motifs" group was significantly higher than that of the other two groups with *p*-values

less than 0.05 by two-tailed Student's t-test (Fig. 4F). This suggests that transcriptional repression by PbAP2-TR is strong in regions containing both binding motifs. For targets with either motif, the average log$_2$(fold change) value was significantly higher in the Motif 1 group than in the Motif 2 group (Fig. 4F). These results verify that Motif 1 is a stronger *cis*-regulatory element for transcriptional repression by PbAP2-TR than Motif 2. Collectively, these results suggest that a single transcription factor can generate variations in the transcript levels of target genes through combinations of different *cis*-regulatory elements. In a previous study, we described a similar mechanism for transcriptional activation by PbSIP2[22]. These *cis*-element dependent variations in the levels of transcriptional activation and repression may be important for the parasite to regulate its complex life cycle with a small number of sequence-specific transcription factors[19].

### PbAP2-TR mainly targets ribosome biogenesis-related genes and exported protein genes and contributes toward establishing their transcription peak patterns

PbAP2-TR targets included 458 genes that were functionally annotated in the PlasmoDB (https://plasmodb.org/plasmo/app)[32] (Supplementary Data 3). To explore deeper insights into the role of transcriptional repression by PbAP2-TR in asexual blood stage development, we classified these annotated genes into characteristic groups. Among 13 groups, the "protein biogenesis" group was the largest, containing 94 target genes (Fig. 5A). These mainly included ribosome-related genes, such as 40S and 60S ribosomal protein genes and nucleolar protein genes. The target genes also included many genes belonging to "metabolism," "exported protein," and "mitochondria" (Fig. 5A). In any of these groups, no significant enrichment of targets associated with Motif 1 or 2 was detected, suggesting that difference in the binding motifs was not related to enrichment of characteristic groups in

PbAP2-TR targets (Fig S7A and Supplementary Data 3). We further performed gene ontology (GO) analysis of the PbAP2-TR targets to explore the enriched gene sets. The analysis revealed that the term "ribosome biogenesis" was most enriched in the target genes with a $p$-value of $2.1 \times 10^{-6}$, consistent with the significant number of ribosome-related genes found in the targets (Supplementary Data 6). Furthermore, several terms related to ribosomes, such as "ribonucleoprotein complex biogenesis," "rRNA processing," and "nucleolus," were included in the 20 most enriched terms (Supplementary Data 6). Other enriched terms included some related to parasite proteins exported into the host cell (such as "host cellular component" and "symbiont-containing vacuole") and two IMC-related terms ("pellicle" and "inner membrane pellicle complex") (Supplementary Data 6).

In the course of the *Plasmodium* IDC, the expression of most genes changes with cell stage progression rather than being constantly expressed, and genes of the same functional group mostly show similar expression patterns[10,14,33]. To investigate whether PbAP2-TR contributes to the establishment of unique expression peak patterns for each functional gene group, we assessed the transcript levels of target-enriched group genes at 4 (ring, before PbAP2-TR expression), 8 (early trophozoite, beginning of PbAP2-TR expression), and 16 hpi (late trophozoite, end of PbAP2-TR expression) (Supplementary Data 7). For most genes with the GO term "ribosome biogenesis" and "nucleolus," their transcript per million (TPM) values peaked at 8 hpi when PbAP2-TR expression began (Figs. 5B and S7B). Similarly, the expression peaks for most *Plasmodium* exported protein genes (including *exp1–3* and *ibis1*) were also observed at 8 hpi (Figs. 5B and S7B). The "pellicle" genes mostly showed no notable change or a slight decrease from 4 to 8 hpi, but similar to ribosome-related genes, their expression decreased towards 16 hpi (Figs. 5B and S7B). Collectively, gene groups enriched in PbAP2-TR targets commonly showed patterns of down-regulation from 8 to 16 hpi, during which PbAP2-TR is expressed. Notably, expression of targets in these groups was mostly upregulated in *pbap2-tr*-DiCre^Rapa+ compared to *pbap2-tr*-DiCre^Rapa- (Fig S7C). Thus, these results suggest that PbAP2-TR contributes to the establishment of transcription peak patterns in its target genes by repressing their transcription during the IDC.

## PbAP2-TR recruits PbMORC as a co-factor

To further investigate the mechanism of transcriptional repression by PbAP2-TR, we performed rapid immunoprecipitation mass spectrometry (MS) of endogenous proteins (RIME), which involves ChIP, followed by MS analysis, and explored its co-factors. ChIP was conducted using PbAP2-TR::GFP^*pbap2g(-)* and WT (as controls) parasites at 12 hpi, as described for the ChIP-seq analyses. IPed proteins were harvested by on-bead digestion with trypsin and subjected to liquid chromatography-tandem MS (LC-MS/MS) analysis. MS data obtained from four biologically independent experiments were compared between PbAP2-TR::GFP^*pbap2-g(-)* and WT. Through this analysis, we identified ten proteins that were unique to PbAP2-TR::GFP^*pbap2-g(-)* ChIP (detected in at least three of four replicates) and one protein that was more than five-fold enriched in PbAP2-TR::GFP^*pbap2-g(-)* ChIP compared to WT ChIP, with a $p$-value less than 0.01 by a two-tailed Student's t-test (Fig. 6A and Supplementary Data 8). Among these, PbAP2-TR itself and PbMORC had average quantitative values (aveQVs) higher than 20, whereas the values for the others were lower than 3, suggesting that PbMORC is a strong candidate for a PbAP2-TR co-factor (Fig. 6A and Supplementary Data 8). This result is consistent with the previous studies in *P. falciparum*, which detected PbAP2-TR ortholog, PfAP2-G5, through the co-IP with PfMORC[34,35].

*Plasmodium* parasites possess a single MORC, which has a conserved GHKL (gyrase, Hsp90, histidine kinase, and MutL)-ATPase domain and several Kelch motifs (Fig. 6B)[34]. To verify that PbMORC is a PbAP2-TR co-factor, we performed the reciprocal RIME analysis at 12 hpi. To conduct this experiment on the *pbap2-g* knockout background

as with the above RIME analysis, a parasite line expressing GFP-fused PbMORC was generated, and *pbap2-g* was disrupted in this parasite [PbMORC::GFP^*pbap2-g(-)*, FigS8A]. The RIME analysis using PbMORC::GFP^*pbap2-g(-)* detected 146 proteins that were unique or more than five-fold enriched (Fig. 6C and Supplementary Data 8). These proteins included PbMORC and PbAP2-TR with the first and third highest aveQVs of 121.3 and 47.5, respectively (Fig. 6C and Supplementary Data 8), which supports the notion that PbMORC acts as a major co-factor of PbAP2-TR. Next, to confirm that PbAP2-TR recruits PbMORC to the upstream regions of target genes as a co-factor, we performed ChIP-seq analysis using PbMORC::GFP^*pbap2-g(-)* at 12 hpi. Duplicate experiments identified 1864 and 1984 peaks, of which 1436 peaks (77% of the experiment 1 peaks) overlapped (Pearson correlation coefficient = 0.94) (Fig. 6D and S8B, and Supplementary Data 9). Among these common peaks, 547 peaks overlapped with those of PbAP2-TR, accounting for 54% (547/1017 peaks) of the PbAP2-TR peaks. Consistently, Motifs 1 and 2 were significantly enriched in the PbMORC peak regions with $p$-values of $1.5 \times 10^{-52}$ and 0.035, respectively, by Fisher's exact test (Supplementary Data 9). Furthermore, the PbMORC ChIP-seq coverage showed clear enrichment patterns around the summits of most PbAP2-TR peaks (Fig. 6E), including those that were classified as non-overlapping with the PbMORC peaks under the thresholds of fold enrichment > 3 and $q$-value < 0.01 in macs2 peak calling (Fig S8C and S8D). These results indicate that PbAP2-TR genome-widely recruits PbMORC to the upstream regions of its target genes.

Importantly, while ChIP-seq showed co-localization in the PbAP2-TR binding regions, only 38% (547/1436 peaks) of the PbMORC peaks overlapped with the PbAP2-TR peaks (Fig. 6F). Motif enrichment analysis of the remaining PbMORC peaks (889 peaks) detected ACACA, TGCAY, and YGTCT, along with their related motifs (Fig. 6G and Supplementary Data 9). These results imply that PbMORC is recruited to the genome not only by PbAP2-TR but also through other factors. In the PbMORC RIME, two other AP2 family members, PBANKA_0521700 and PbAP2-P (PBANKA_0939100), were identified with the second and fourth highest aveQVs of 69.3 and 27.8, respectively (Fig. 6C and Supplementary Data 8) while they were not detected in the PbAP2-TR RIME (Supplementary Data 8). These findings may suggest that PbMORC functions as a co-factor of PBANKA_0521700 and PbAP2-P during the IDC, independently of PbAP2-TR. Notably, in *P. falciparum*, the orthologs of these transcription factors, PF3D7_0420300 and PfAP2-P (PF3D7_1107800), were also detected in the co-IP with PfMORC[34,35]. In addition, PF3D7_0420300 and PfAP2-P have been reported to bind to ACACA and TGCATG, respectively[31,36], which were analogous to those enriched in the PbMORC peaks.

RIME analyses of PbAP2-TR and PbMORC additionally identified several putative transcriptional regulators. The proteins commonly detected in the two RIME analyses included EELM2 (PBANKA_1234600) and ISWI (PBANKA_1123500) (Figs. 6A, C, Supplementary Data 8). Furthermore, other putative transcriptional regulators, such as CHD1 (PBANKA_0907200), ELM2 (PBANKA_0205000), and HDAC1 (PBANKA_0826500), were uniquely identified in the PbMORC RIME (Fig. 6C and Supplementary Data 8). While their aveQVs in the RIME were relatively low compared to those of PbAP2-TR and PbMORC, these proteins might also contribute to the transcriptional regulation during the IDC as potential interaction partners of PbAP2-TR and/or PbMORC.

## Discussion

In *Plasmodium*, transcriptional repressors have been shown to play a role in regulating cell fate bifurcations, namely sexual differentiation and sex determination, and in silencing subtelomeric multigene family expression[36–42]. These processes are regulated in a mutually exclusive or clonally variant manner and often involve trimethylated histone H3 Lys9, which is a hallmark of heterochromatin[43–45]. In contrast, asexual

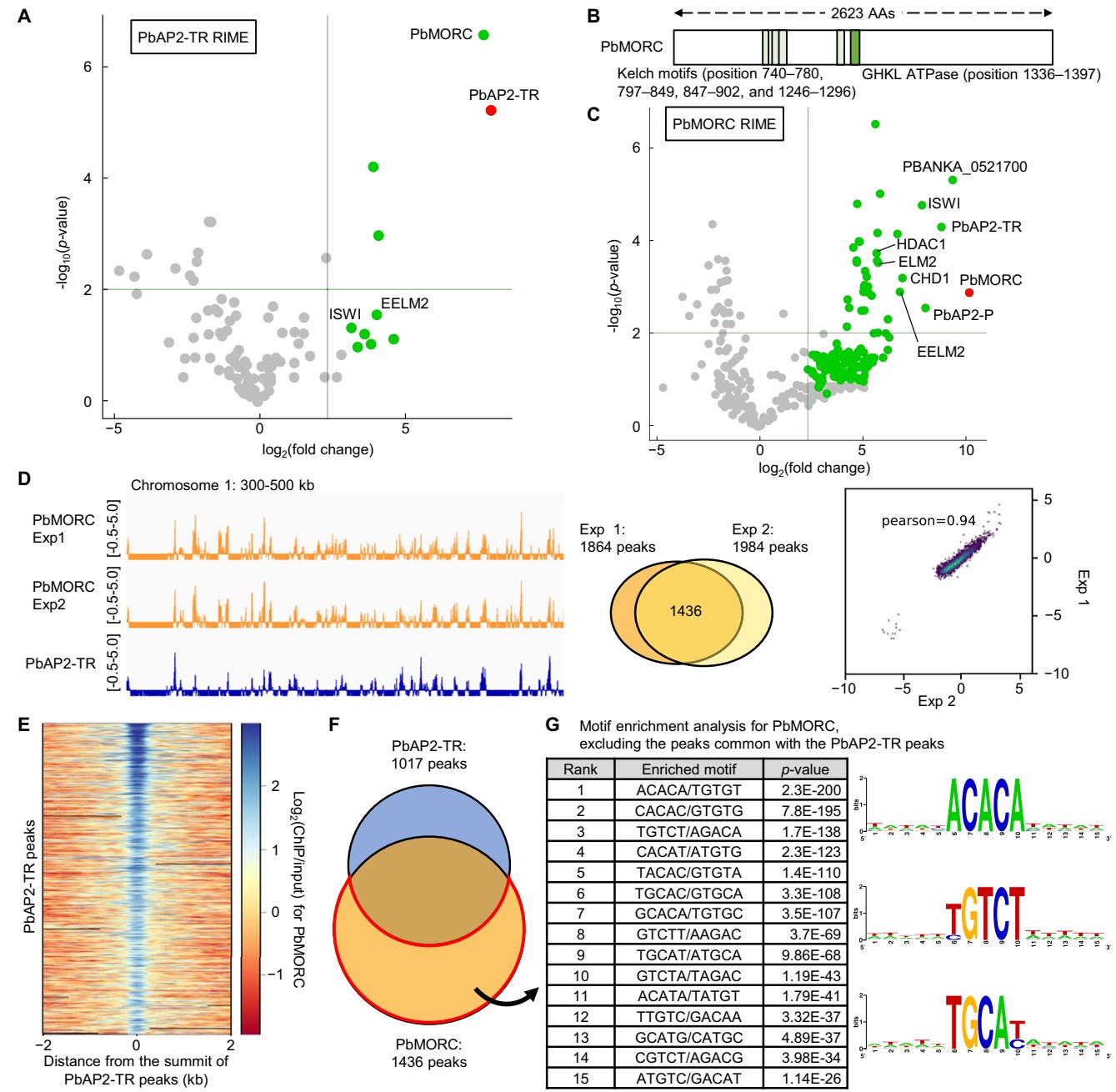

**Fig. 6 | Recruitment of PbMORC by PbAP2-TR as a co-factor. A** Volcano plot showing proteins enriched in PbAP2-TR::GFP RIME relative to wild-type (WT) at 12 hpi. Four biologically independent experiments were performed. Proteins uniquely identified in PbAP2-TR::GFP samples (≥3 of 4 replicates) are indicated by green dots. For proteins with an average quantitative value of 0, pseudo fold change values were calculated by adding a pseudo-count value (0.1). *p*-values were calculated using a two-tailed Student's t-test. Horizontal and vertical lines indicate *p*-value of 0.01 and fold change of 5, respectively. **B** Schematic of PbMORC. GHKL (gyrase, Hsp90, histidine kinase, and MutL)-ATPase and Kelch motifs are indicated by green and light green, respectively. **C** Volcano plot showing proteins enriched in RIME of PbMORC::GFP relative to WT at 12 hpi. Four biologically independent experiments were performed. Proteins uniquely identified in PbMORC::GFP samples (≥3 of 4 replicates) are

indicated by green dots. *p*-values were calculated using a two-tailed Student's t-test. **D** Integrative Genomics Viewer images for PbMORC ChIP-seq experiments 1 and 2 on chromosome 1, with PbAP2-TR ChIP-seq experiment 1 shown below. Histograms show log₂-transformed ChIP/input ratio (bin size = 10 bp). Scales are indicated in square brackets. Venn diagram indicates overlapping peaks between two PbMORC experiments. Scatter plot compares log₂(ChIP/input) values across replicates at 1-kb bin. **E** Heatmap showing log₂(ChIP/input) of PbMORC at PbAP2-TR peaks. Peak regions are aligned in ascending order of their *q*-values. **F** Venn diagram showing the number of overlapping peaks between PbAP2-TR ChIP-seq and PbMORC ChIP-seq. **G** DNA motifs enriched in PbMORC peak regions that did not overlap with PbAP2-TR peaks. *p*-values were calculated using Fisher's exact test. Logos for enriched motifs were generated using WebLogo and are displayed on the right.

blood stage development is a simple replication cycle that proceeds without cell fate diversion. Thus, it could be considered that transcriptional regulation during the IDC does not require transcriptional repressors and proceeds via a cascade of transcriptional activators alone that induce each stage-specific transcription in a "just-in-time"

manner. However, in this study, we demonstrated that the transcriptional repressor PbAP2-TR is essential for the progression of asexual blood stage development. The target genes of PbAP2-TR include many genes related to protein biosynthesis, mainly ribosome biogenesis, and exported protein genes. Ribosome biogenesis is the first step in

protein biosynthesis[46], and exported proteins are expressed to modify host RBCs, which is an important first step for intraerythrocytic development[47,48]. The transcription pattern of these genes peaks during the ring to trophozoite development and decreases towards the late trophozoite stage as their requirement for the later cell stage decreases[33,49]. Our results indicate that PbAP2-TR establishes this transcription peak pattern by repressing their transcription in the course of early to late trophozoite development, thereby being essential for promoting transition from the cell growth to replication phases of the IDC. This role of PbAP2-TR suggests that the "just-in-time" gene expression patterns during IDC are strictly controlled at the transcriptional level through a combination of transcriptional activators and repressors, providing an insight into the role of transcriptional repressors in asexual blood stage development.

In the PbAP2-TR targets, we also detected enrichment of pellicle/IMC-related genes. The pellicle and IMC structures are important for merozoite formation and motility[50–52], and genes related to these structures are included among the targets of PbSIP2, a master transcription factor that regulates merozoite formation[22]. Consistently, their expression in the IDC reaches a peak during schizont development and is downregulated after RBC invasion[14]. Given this, the role of PbAP2-TR is not limited to the establishment of transcription peak patterns through repressing genes that are highly transcribed during early trophozoite development; it also represses genes that are not required for or are possibly harmful to trophozoite development. In a previous study, we revealed that the female transcriptional repressor PbAP2-FG2 plays multiple roles in promoting female development, such as repression of early gametocyte genes and suppression of male fate[39]. These suggest that in *Plasmodium*, transcriptional repressors target wide-variety of genes to play multiple roles, unlike transcriptional activators, whose function is limited to regulate certain stage-specific genes.

MORC family proteins are conserved in eukaryotes, including apicomplexan parasites, and play roles in gene silencing and chromatin compaction[53]. *Plasmodium* parasites have a single MORC ortholog that is essential for asexual blood stage development and is proposed to be involved in chromatin organization and heterochromatin-related gene silencing[34,35]. Besides this broad-scale chromatin regulation, the role of MORC has also been suggested as a transcriptional regulator associated with AP2 transcription factors; *i.e.*, MORC was also identified through co-IP with some AP2 transcription factors, and co-IP with PfMORC and our PbMORC RIME detected several other AP2 transcription factors[34,35,38,39]. Such interactions between the MORC protein and AP2 transcription factors have also been detected in other apicomplexan parasite, *Toxoplasma*[54–58]. In this study, we performed a ChIP-seq analysis of PbMORC and demonstrated that PbMORC is genome-widely recruited to a specific *cis*-regulatory element by PbAP2-TR. These results strongly indicate that MORC is involved as a co-factor in the transcriptional regulation of genes downstream of transcription factor binding sites.

In *Toxoplasma*, MORC functions in a complex with histone deacetylase 3 (HDAC3) to regulate sexual commitment, suggesting that MORC functions are related to histone modifications[54]. Comparably, HDAC1 was detected as an interaction partner of both PbMORC and PfMORC[34,35], suggesting that the cooperation of MORC proteins with histone deacetylases is common in Apicomplexa. In contrast, RIME analysis with PbAP2-TR detected PbMORC as a major co-factor, but no histone-modifying enzymes. Similarly, our previous study revealed PbMORC as the only major co-factor of the female-specific transcriptional repressor complex, PbAP2-FG2/PbAP2R-2[39]. These imply that, in *Plasmodium*, the components of co-factor complexes differ depending on the transcription factor that recruits them; *i.e.* when PbMORC is recruited by PbAP2-TR or PbAP2-FG2, they may repress target gene expression simply by remodeling chromatin states, without the involvement of histone modification. Alternatively, the PbAP2-TR and

PbAP2-FG2 RIME did not detect histone modifying enzymes perhaps because they may initially act in a complex with MORC and subsequently dissociate from genomic DNA while MORC remains associated with chromatin. Following this dissociation, MORC may then recruit histone-modifying enzymes to further modulate the downstream gene expression.

In *P. falciparum*, the role of PbAP2-TR-ortholog, PfAP2-G5, was suggested as suppression of gametocyte fate by repressing the expression of PfAP2-G, a transcription factor essential for triggering gametocyte differentiation[30]. Accordingly, the marked reduction of asexual growth observed upon disruption of *pfap2-g5* was interpreted as a consequence of an increased proportion of parasite population that commits to sexual fate. By contrast, the present study revealed that PbAP2-TR is not involved in gametocytogenesis but regulates gene expression in trophozoites, thereby playing an essential role in asexual proliferation. These findings suggest that PbAP2-TR and PfAP2-G5 have distinct biological roles in *P. berghei* and *P. falciparum*, respectively. Nevertheless, this might be attributed to the fact that in *P. falciparum*, its role in the asexual blood stage development has not been evaluated. In both *P. falciparum* and *P. berghei*, *ap2-g*-knockout parasites outgrow WT parasites, indicating that the *ap2-g*-disruption causes a substantial increase in asexual growth rate owing to the loss of parasite population that differentiates into gametocyte[59,60]. However, the additional knockout of *pfap2-g* in the *pfap2-g5*-knockout parasites could not completely complement the parasite growth rate, which was significantly decreased in the *pfap2-g5*-knockout parasites, to that of WT. Therefore, this result demonstrates that PfAP2-G5 also plays a role in the asexual development. We consider that further investigation of PfAP2-G5 functions during the IDC is important for understanding the transcriptional regulation mechanism that promotes the *Plasmodium* asexual blood stage development.

In summary, our results indicate that transcriptional repression is essential for establishing precise transcriptional profiles during *Plasmodium* asexual blood stage development. In the IDC transcriptomes, several gene groups show different peak patterns of gene transcription, which constitutes the cascade of "just-in-time" transcription, suggesting that multiple transcriptional repressors are involved in generation of this cascade. *Plasmodium* parasites have several AP2 transcription factors essential for asexual blood stage development[24,26], some of which have not been functionally investigated. As our ChIP-seq of PbMORC detected the enrichment of several motifs other than PbAP2-TR-binding motifs, these factors may include other transcriptional repressors that recruit PbMORC. We believe that further exploring the functions of transcription repressors is important for understanding the processes of parasitic blood stage development.

## Methods

### Ethics statement
All experiments were performed in accordance with the recommendations of the Guide for the Care and Use of Laboratory Animals of the National Institutes of Health to minimize animal suffering and were approved by the Animal Research Ethics Committee of Mie University (permit number 23–29).

### Parasite preparation
The parasites were inoculated into female ddY mice (*Mus musculus*, an outbred strain purchased from Japan SLC Inc., 5–6-week-old), which were kept in a temperature-controlled room (22–24 °C) under a 16:8 h light/dark cycle with relative humidity within 40–60%. All parasites used in this study were derived from the wild-type (WT) *P. berghei* ANKA strain. Transgenic parasites generated using the CRISPR/Cas9 method were derived from the WT-originated Cas9-expressing parasite, PbCas9, which has a Cas9 expression cassette at the *p230p* locus[39]. The *pbap2-tr*-DiCre parasite was generated from PbCas9<sup>DiCre</sup>, which

constitutively expresses Cre59 (Thr19–Asn59) fused with FKBP12 and Cre60 (Asn60–Asp343) fused with FRB in addition to Cas9[22].

In vitro cultures of whole blood from infected mice were performed using RPMI1640 medium supplemented with 25% fetal calf serum and penicillin/streptomycin at 37 °C in 5% $CO_2$ and 10% $O_2$. For cell cycle synchronization, infected blood was cultured for 16 h, and mature schizonts produced in the cultures were harvested by density gradient centrifugation using an iodixanol solution (Optiprep), whose density was adjusted to 1.077 g/mL by mixing with tricine solution. The harvested schizonts were then intravenously injected into mice. Parasitemia and parasite morphology were assessed on Giemsa-stained blood smears.

### Generation of transgenic parasites via the CRISPR/Cas9 system
For gene editing with the CRISPR/Cas9 system using PbCas9 and PbCas9[DiCre], donor DNAs and single guide RNA (sgRNA) vectors were constructed according to the previously developed protocol[27]. Briefly, templates for donor DNAs were constructed using overlap PCR and cloned into pBluescript KS (+) by In-Fusion cloning (Takara) at the XhoI and BamHI sites. Donor DNAs were then amplified by PCR from the constructed plasmid. The target sequences of the sgRNAs were designed using CHOPCHOP. sgRNA templates were constructed by annealing DNA oligos and were cloned into the sgRNA vector using the DNA Ligation Kit (Takara).

Transfection was performed using cultured schizonts and DNA constructs described above. Briefly, recipient parasites were cultured for 16 h, and schizonts were harvested by density gradient centrifugation. The harvested schizonts were then washed once with RPMI1640, transfected with DNA constructs using the Amaxa Basic Parasite Nucleofector Kit 2 (LONZA), and immediately inoculated into mice. All transfectants were selected by treating the infected mice with 70 μg/mL pyrimethamine in their drinking water from 30 h after transfection. For CRISPR/Cas9 gene editing, recombination was confirmed using PCR and/or Sanger sequencing, and clonal parasites were obtained by limiting dilution. All the primers used in this study are listed in Supplementary Data 10.

### Fluorescence microscopic analysis
Fluorescence analysis was performed using the Olympus BX51 microscope with Olympus DP74 camera. For nucleus detection, parasites were stained with 1 ng/mL Hoechst 33342 for 10 min at 37 °C.

### Conditional knockout of *pbap2-tr*
Whole blood from mice infected with *pbap2-tr*-DiCre (or *pbap2-tr*-DiCre[pbap2-g(-)]) parasites was cultured. The cultures were then split into two parts; a 3/20 volume of rapamycin (Wako) solution (200 μM DMSO stock) was added to one part (final concentration of 30 nM), and the same volume of DMSO was added to the other as a control. The parasites were cultured for 16 h, and schizonts were harvested and inoculated into mice. For in vitro phenotype analysis, the parasites were cultured again at 10 hpi. For in vivo phenotype analysis, parasites were grown in mice for 3 days, and parasitemia was assessed daily by Giemsa staining. Genotyping PCR to confirm recombination at the *pbap2-tr* locus was performed at 16 h after starting the cultures before schizont harvest and on day 3 of the in vivo phenotype analysis. The primers used for genotyping are listed in Supplementary Data 10.

### ChIP-seq and sequencing data analysis
ChIP-seq was performed without using spike-in normalization. Infected blood was passed through a Plasmodipur filter, and formalin solution was immediately added to a final concentration of 1% for cell fixation. After 1 h of fixing at 30 °C, RBCs were lysed in ice-cold 1.5 M $NH_4Cl$ solution, and residual cells were lysed in SDS lysis buffer (50 mM Tris-HCl, 1% SDS, 10 mM EDTA). The cell lysate was sonicated using the Bioruptor (Cosmo Bio), and immunoprecipitation was performed at

4 °C using anti-GFP polyclonal antibodies (Abcam, ab290, Rabbit polyclonal, reactive against all variants of *Aequorea victoria* GFP, GR19413-1, 1:250) conjugated to Protein A Magnetic Beads (Invitrogen). After overnight incubation with rotation, the beads were washed five times with low-salt wash buffer (20 mM Tris-HCl, 0.1% SDS, 2 mM EDTA, 1% Triton X, 150 mM NaCl) and three times with high-salt wash buffer (20 mM Tris-HCl, 0.1% SDS, 2 mM EDTA, 1% Triton X, 500 mM NaCl). The beads were then resuspended in extraction buffer (10 mM Tris-HCl, 1% SDS, 5 mM EDTA, 300 mM NaCl) and were incubated for 15 min at room temperature to extract the immunoprecipitated chromatin. DNA fragments were purified from the chromatins after treatment with RNase A and proteinase K. Libraries were constructed from the DNA fragments using the KAPA HyperPrep Kit (Kapa Biosystems). Next-generation sequencing (NGS) was performed using the MGI DNBSEQ-G400. DNA fragments purified from cell lysates before immunoprecipitation were also sequenced as input sequence data. Two biologically independent experiments were performed for each ChIP-seq experiment.

Reads in the sequence data were mapped onto the reference genome sequence of *P. berghei* (v3.0, downloaded from PlasmoDB) using Bowtie 2. Those aligned onto more than two sites in the genome were removed from the mapping data, and peaks were called by the macs2 callpeak using the input sequence data as control (fold enrichment > 3.0, *q*-value < 0.01, and setting --call-summit option on). The ChIP-seq enrichment on the genome was visualized as raw read coverage normalized by library size using bamCoverage (-bs 10, --normalizeUsing RPKM) or $\log_2$-transformed ChIP/input ratio using bamCompare (-bs 10). The replicate data were compared using IDR1D (https://idr2d.mit.edu/)[61] (max gap = 100) and multiBigwigSummary bins (-bs 1000) to assess their reproducibility. Fraction of reads in peaks (FRiP) was calculated using plotEnrichment, and normalized strand cross-correlation coefficient (NSC) and relative strand cross-correlation coefficient (RSC) were assessed using phantompeakqualtools for quality check of each ChIP and input data. The parameters for all programs were set as default unless otherwise indicated. Common peaks in duplicates were defined as those with a distance less than 150 bp between their peak summits.

The enrichment of motifs within 50 bp of the peak summits was analyzed using Fisher's exact test. For this motif enrichment analysis, peaks located within 1 kb of the end of each chromosome were excluded to avoid detection of the telomeric repeat sequence TTYAGGG (Y = T or C). In the *P. berghei* reference genome, 13 of 28 chromosome end sequences are fully resolved, and these include less than 1-kb telomeric repeat sequences. Repeats of TTYAGGG are also found within subtelomeric 2.3-kb repeat units. Genes with peaks within 1200 bp of their start codons were identified as target genes. The 1200-bp threshold was determined based on the previous observations that binding motifs of AP2-O and AP2-Sp were detected within 1200 bp from the start codons of the major ookinete and sporozoite-specific genes, respectively[62,63]. The gene ontology analysis was performed using GSEABase and GOstats programs.

### DIP-seq analysis
DIP-seq was performed as previously described[64]. Briefly, the sequences for the AP2 domains of PbAP2-TR (164–336 for AP2 domains 1–2 and 1458–1527 for AP2 domain 3) were cloned into the MBP fusion vector pMal-c5X (NEB). *E. coli* transformed with the plasmids was cultured for 12 h at 37 °C, and the expression of MBP-fused proteins was then induced with isopropyl β-D-thiogalactopyranoside (Wako, final concentration of 200 nM). Recombinant AP2 domains fused with MBP were purified using amylose resin (NEB) and were mixed with *P. berghei* ANKA genomic DNA fragments. After 30 min of incubation at room temperature, the recombinant proteins and bound DNA fragments were purified using amylose resin. DNA fragments were then subjected to library preparation and NGS using the Illumina NextSeq 500, as

described for the ChIP-seq analysis. Genomic DNA fragments were sequenced as inputs before use in the DIP. Sequence data were analyzed similar to that in ChIP-seq, except for the parameters of peak calling (fold enrichment > 2.5, $q$-value < 0.01). Motif enrichment analyses were performed separately for three chromosome sets (chromosomes 1–8, 9–12, and 13–14) because the analysis using all DIP-seq peaks detected several motifs with $p$-values less than $5.0 \times 10^{-324}$, which is the smallest positive real number on the R platform.

### RNA-seq and sequence data analysis

For RNA-seq analysis using *pbap2-g*-knockout parasites, parasite cell cycles were synchronized as described in the parasite preparation. For RNA-seq analysis using *pbap2-tr*-DiCre, rapamycin-dependent disruption of *pbap2-tr* was induced, and schizonts of *pbap2-tr*-DiCre[Rapa-] and *pbap2-tr*-DiCre[Rapa+] were inoculated into mice, as described for the conditional knockout of *pbap2-tr*. Whole blood was harvested at 4, 8, and 16 hpi for *pbap2-g*-knockout parasites and at 12 hpi for *pbap2-tr*-DiCre[Rapa-] and *pbap2-tr*-DiCre[Rapa+] parasites. The blood was then passed through a Plasmodipur filter. RBCs were lysed in ice-cold 1.5 M NH$_4$Cl solution, and total RNA was extracted from the residual parasites using the Isogen II reagent (Nippon Gene). RNA-seq libraries were prepared using the KAPA mRNA HyperPrep Kit (Kapa Biosystems) and were sequenced using the MGI DNBSEQ-G400. Three biologically independent experiments were performed for each sample. The sequence data were mapped onto the reference *P. berghei* ANKA genome sequence using HISAT2, with the maximum intron length set as 1000. The number of reads mapped to each gene was calculated using featureCounts and compared between samples using DESeq2. To evaluate the replicate consistency, principal component analysis plots, sample-to-sample distance heatmaps, and dispersion plots were produced using DESeq2 plotPCA, dist, and plotDispEsts, respectively. For differential expression analyses, DESeq2 apeglm shrinkage[65] was applied to the log$_2$(fold change) to accurately evaluate the effect sizes without arbitrarily filtering out genes with low counts. Subtelomeric multigene families (*pir* and *fam*) were removed from the differential expression analysis. Default parameters were set for all programs unless indicated otherwise.

### Dual-luciferase reporter assay

Plasmids were constructed to generate luciferase reporter lines as previously described[37]. Briefly, the upstream regions of *dlat* and *bpl-1* were amplified from *P. berghei* genomic DNA by PCR and inserted upstream of the firefly luciferase gene in the reporter plasmid pCen-luc using the *Kpn*I and *Nhe*I sites. In addition to the firefly luciferase gene, pCen-luc contains the Renilla luciferase gene expression cassette as an internal control and the human *dhfr* gene as a drug-selectable marker. The expression of these two genes is under the control of the bidirectional *P. berghei* elongation factor 1-alpha gene promoter. The plasmid also contains the centromere sequence from the *P. berghei* chromosome 5[66]. Mutations within *dlat* and *bpl-1* promoters were introduced using overlap PCR before integration of their upstream sequences into the reporter plasmid.

Parasites carrying pCen-luc were synchronized as described in the parasite preparation method and were harvested from the tail vein of infected mice at 14 hpi. RBCs were lysed in ice-cold lysis solution (1.5 M NH$_4$Cl, 10 mM EDTA, and 0.1 M KHCO$_3$) for 5 min, and the residual cells were subjected to dual-luciferase reporter assays using the Dual-Luciferase Reporter Assay System (Promega). The assays were performed according to the manufacturer's instructions. Firefly and *Renilla* luciferase expression was assessed using the GloMax 96 Microplate Luminometer (Promega). The assays were performed in biological triplicate for each sample.

### RIME

ChIP was performed as described for ChIP-seq analysis. The beads with immunoprecipitated chromatins were washed twice with 100 mM ammonium hydrogen carbonate (AMBIC) solution. Proteins on the washed beads were digested with 10 μl of trypsin (Promega) in 100 mM AMBIC at an enzyme-to-protein ratio of 1:100 (wt/wt) for overnight at 37 °C and were further incubated for 4 h at 37 °C with additional 10 μl of trypsin. The peptides released from the beads through digestion were then purified using a C18 tip (GL-Science, Tokyo, Japan) and then subjected to nanocapillary reversed-phase LC-MS/MS analysis using a C18 column (12 cm × 75 μm, 1.9 μm, Nikkyo Technos, Tokyo, Japan) on a nanoLC system (Bruker Daltoniks, Bremen, Germany) connected to a timsTOF Pro mass spectrometer (Bruker Daltoniks) and a modified nano-electrospray ion source (CaptiveSpray; Bruker Daltoniks). From the MS/MS data, IPed proteins were identified using DataAnalysis version 5.2 (Bruker Daltoniks) and MASCOT version 2.7.0 (Matrix Science, London, UK) against the Uniprot_Plasmodium_berghei_ANKA_strain database (4948 sequences; 3412795 residues). Protease specificity was set for trypsin (C-term, KR; Restrict, P; Independent, no; Semispecific, no; two missed and/or nonspecific cleavages permitted); conversion of the N-terminal Gln to pyro-Glu and oxidation of methionine were considered as possible modifications. The mass tolerance for precursor ions and fragment ions were ±15 ppm and ±0.05 Da, respectively. The threshold score/expectation value for accepting individual spectra was $P < 0.05$. Quantitative values were determined using Scaffold5 version 5.1.2 (Proteome Software, Portland, OR, USA). Four biologically independent experiments were performed for each sample. Fold change and $p$-value (two-tailed Student's t-tests) were calculated using Microsoft Excel.

### Reporting summary

Further information on research design is available in the Nature Portfolio Reporting Summary linked to this article.

## Data availability

All ChIP-sequencing, RNA-sequencing and DIP-sequencing data generated in this study have been deposited in the Gene Expression Omnibus database under accession numbers GSE290541, GSE290542, GSE290544, and GSE290545. All MS/MS data for RIME have been deposited in the ProteomeXchange Consortium under dataset PXD067426. Source data are provided with this paper.

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

## Acknowledgements

This work was supported by the Japan Agency for Medical Research and Development (253fa627002h to S.I.), the Japan Society for the Promotion of Science (24K10187 to TN; 23H02709 to YM; 23K06515 to IK), and the Grant for Joint Research Project of the Research Institute for Microbial Diseases, The University of Osaka (JRPRIMD25B9 to IK). We would like to express our gratitude to Dr. Hiroko Kato and Dr. Ryohei Narumi, Central Instrumentation Laboratory, Research Institute for Microbial Diseases, The University of Osaka for performing analysis using timsTOF Pro and providing insight and expertise that greatly assisted the research.

## Author contributions

T.N. and Y.M. designed the study. T.N. performed all experiments, except for next generation sequencing and liquid chromatography-tandem mass spectrometry analysis. These analyses were performed by DNAFORM and Central Instrumentation Laboratory, Research Institute for Microbial Diseases, The University of Osaka, respectively. T.N. performed all formal data analyses and statistical analyses. I.K. contributed to the generation of transgenic parasites. S.I. contributed technical assistance/suggestions. All obtained funding for the study. Y.M. supervised the project. T.N. and Y.M. wrote the paper.

## Competing interests

The authors declare no competing interests.
