## [Transparent Peer Review file · Nature Communications]

Precise gene regulation through transcriptional repression is essential for *Plasmodium berghei* asexual blood stage development

Corresponding Author: Professor Masao Yuda

Version 0:

Reviewer comments:

Reviewer #1

(Remarks to the Author)

This manuscript describes the function of the *Plasmodium berghei* transcription factor the authors name Pb-TR. The authors convincingly show that this AP2 domain-containing protein is essential for parasite development during the asexual replication cycle through a conditional KO. They go on to analyze its targets in the genome through both ChIP-Seq and DIP-Seq and convincingly identify candidate motifs. Interestingly, two independent motifs are identified and specify either the first two AP2 domains or the last single AP2 domain. The authors go on to determine that Pb-TR is more associated with genes identified in the ChIP-Seq analysis that are markers of early stages rather than late stages, leading to the hypothesis that Pb-TR is a repressor. They then further establish a phenotype using their conditional KO RNA-Seq, and show in a luciferase assay that the motifs identified can lead to higher reporter luciferase bioluminescence when mutated. Finally, they identify MORC as an interactor using both MS and by comparing ChIP-Seq profiles. The data is overall compelling, and this is a useful addition to the understanding of transcription in the rodent model of malaria.

There are a some limitations of the study.

- Given the strength of the phenotype, I found it surprising that the transcriptional changes in the RNA-Seq data following excision were somewhat limited and wondered whether 12 hpi was indeed the right time point to conduct this assay. Only a minute number of genes with peaks are found to be upregulated. Is this low proportion expected, and is it sufficient to conclude that Pb-TR is a transcriptional repressor?
- Some of the analysis and the interpretations in the paper could be deeper. For instance, it would be interesting to dissect and discuss further the contribution of the different AP2 domains and whether they may be associated with different targets. The discussion around the functional enrichment of Pb-TR targets is perhaps not deserving of a main figure, as it doesn't really enhance the understanding of the phenotype, and no attempt is made to use the RNA-Seq in tandem. Another example is the other proteins identified by RIME, which include several chromatin-associated proteins, but no attempt is made at discussing whether they may also be present at Pb-TR sites and how that could play into the transcriptional regulation.
- An interrogation arising from the paper is the contrast with the phenotype of the *Plasmodium falciparum* ortholog of Pb-TR. The authors contrast this in the discussion, but this could be really enhanced; a divergent role of an AP2 in different spp. is an interesting observation, and the authors could do more comparative analysis to establish whether there is a possible divergence. For instance, comparing targets of the APG5 and Pb-TR may be an interesting first step. Truly establishing a divergence between the two species would really enhance the impact of the paper.

Minor points:

A citation of a VEuPathDB publication should be included.

Reviewer #2

(Remarks to the Author)

Summary: The study identifies PbAP2-TR (Trophozoite Repressor) as a trophozoite-stage AP2 transcription factor that functions as a transcriptional repressor essential for *Plasmodium berghei* asexual blood-stage development. Using DiCre conditional knockout, ChIP-seq/DIP-seq, time-course RNA-seq, motif mutagenesis luciferase reporters, and RIME/ChIP-seq

for PbMORC, the authors show that PbAP2-TR binds two motifs (GTTGY and GTTCT) via distinct AP2 domains, represses target genes (notably ribosome biogenesis and exported proteins) from 8-16 hpi, and recruits PbMORC as a co-factor; only ~38% of MORC peaks overlap with PbAP2-TR, implying additional recruiters.

Originality and field context: Prior work in *Toxoplasma gondii* established a MORC–HDAC3–ApiAP2 repression axis, and recent *P. falciparum* studies show PfMORC is essential, associates with heterochromatin, and co-IPs with multiple ApiAP2 proteins (including PfAP2-G5). The present study moves beyond association to a cis-encoded, stage-bounded mechanism in malaria parasites: a specific trophozoite ApiAP2 repressor (PbAP2-TR) with dual-motif/domain logic that recruits PbMORC to enforce a precise 8-16 hpi repression program. Notably, while PfAP2-G5 co-purifies with PfMORC, it is non-essential across the IDC and primarily linked to restraining gametocyte commitment, whereas PbAP2-TR is essential for asexual progression—underscoring a mechanistic advance rather than a mere extension of Pf co-IP observations. In short, the present story extends the *Toxoplasma* paradigm and fills a key mechanistic gap left by the PfMORC literature, making it a substantive, field-moving advance rather than an incremental confirmation.

Major strengths

- Clear stage specificity and genetic essentiality. PbAP2-TR is expressed in trophozoites and DiCre excision causes a trophozoite-stage developmental arrest with careful staging controls.
- Cohesive multi-omics mechanism. ChIP-seq and DIP-seq define two motifs and assign them to different AP2 domains; luciferase mutagenesis shows cis-repressive function with Motif 1 > Motif 2 effect sizes.
- Biological coherence. Target-enriched functional categories (ribosome biogenesis, exported proteins, pellicle/IMC) decline from 8-16 hpi, matching the proposed repressive window.

A) Key concerns about Transcriptomics (analysis & visualization)

The transcriptomic component is central to your mechanistic claim and is, overall, well integrated with the genetics and ChIP data. The authors analyze (i) an 8-16 hpi progression in an ap2-g- background to align with the trophozoite window of PbAP2-TR expression, and (ii) an acute pbap2-tr DiCre ± rapamycin contrast at 12 hpi with three biological replicates, mapped with HISAT2/featureCounts and modeled in DESeq2—good choices that match the biology and enable clear cross-validation between time course and inducible loss of function. At present, results are primarily visualized with volcano plots (Fig. 3A, Fig. 4A), and gene-set summaries are shown as TPM-scaled line plots across 4/8/16 hpi (Fig. 5B).

- Use MA plots with shrunken log₂FC values instead of volcano plots. The current volcanoes (Figs. 3A and 4A) overemphasize extreme p-values, which hides the role of expression strength and gives undue weight to low-count, highly significant genes. MA plots (mean expression vs shrunken log₂FC, e.g. with DESeq2 apeglm/ashr) make effect sizes at biologically relevant expression levels clearer. Please re-render Figs. 3A and 4A as MA plots, marking FDR and |LFC| cutoffs directly; a simplified volcano can remain in the supplement if desired.
- Show full QC and replicate information. For the 12 hpi DiCre contrast (n = 3 per condition), include in the Supplement a PCA plot, a sample-to-sample distance heatmap, and the standard DESeq2 dispersion–mean plot to document replicate consistency and treatment separation. Clearly state the number of replicates for the 8/16 hpi time points and provide the same QC panels, so readers can be confident that the time-course results are robust.
- Clarify the gene-set expression patterns. Fig. 5B's line plots (TPM normalized to 4 hpi) are informative but overcrowded. Replace or supplement them with row-z-scored heatmaps of the main gene sets (“ribosome biogenesis,” “nucleolus,” “exported proteins/pellicle”) across 4/8/16 hpi, annotated with PbAP2-TR target status and Motif 1/Motif 2 presence near the TSS. This format will make the coordinated 8–16 hpi down-regulation immediately visible and directly link expression changes to cis-regulatory logic.

B) Key concerns about MORC/AP2-TR

- In Fig. 6C, the IGV snapshots don't convincingly demonstrate co-localization: the two PbMORC replicates are shown on chromosome 1 with their own y-axis scales, and the PbAP2-TR track is plotted separately below, making direct, like-for-like visual comparison difficult; several MORC peaks appear without an obvious AP2-TR peak in the same view. This impression actually matches your own quantification that only 38% of MORC peaks overlap AP2-TR (Fig. 6E), so the panel, as drawn, tends to undercut the text's claim of “genome-wide co-localization.” Consider (i) replotting matched loci with identical y-axis scaling and a single, aligned multi-track view (MORC rep1/rep2 and AP2-TR together), (ii) adding zoom-ins on representative overlap and non-overlap sites, and (iii) emphasizing the more appropriate summary evidence (the heat map of MORC signal centered on AP2-TR peaks in Fig. 6D and the 38% overlap in Fig. 6E) while softening the wording from “genome-wide co-localization” to “co-localized at a subset of AP2-TR sites.”
- To move beyond correlative evidence of MORC co-recruitment, perform reciprocal co-IP or BioID to confirm direct complex formation and reinforce the central model.

C) Minor comments

- Your RIME list (Table S8) includes a protein bearing an ELM2 domain—precisely the class of ELM2–SANT co-repressors that scaffold MORC–HDAC3 complexes in *Toxoplasma*—yet the manuscript concludes that no chromatin regulators were detected and emphasizes MORC acting alone; please acknowledge this candidate and, ideally, test its association/function (reciprocal IP).

- ChIP-seq robustness: strengthen the ChIP-seq methods and reporting by providing replicate concordance and quality metrics (IDR, FRiP, NSC, RSC) for both AP2-TR and MORC datasets, clarifying how peaks were called and thresholds defined. Explicitly justify the use of the ± 1200 bp promoter window in *P. berghei*, state whether spike-in normalization was applied, and include antibody validation details (lot, specificity) for the GFP ChIP. Finally, define the exact criteria for calling “targets” directly in the main text, not only in tables.

D) Optional but valuable additions

- Consider performing ATAC-seq in the PbAP2-TR KD context to directly link transcriptional repression with chromatin compaction at target loci.

Reviewer #3

(Remarks to the Author)

The authors described the function of PbAP2-TR, an AP2-domain containing transcriptional factor, in the regulation of malaria parasite asexual development. When this gene was conditionally knocked out, parasite growth was arrested at the trophozoite stage. ChIP-seq analysis revealed PbAP2-TR-regulated genes and their binding motifs. Differential expression analysis by RNA-seq indicated that PbAP2-TR is a transcriptional repressor, and this conclusion was confirmed by an episomal expression cassette with PbAP2-TR binding motifs. ChIP followed by mass spectrometry analysis revealed that PbMORC was associated with PbAP2-TR.

This work revealed the novel functions of PbAP2-TR, which is significant in the field. Authors put a great effort to provide detailed information by advanced technologies such as Dicer-based conditional KO, ChIP-seq, and RNA-seq. The overall study is excellent and innovative. However, authors need to address quite a few questions to completely cover the whole story of this gene.

1. First of all, this study revealed the potential hidden function of PfAP2-G5, the ortholog of PbAP2-TR in *P. falciparum*. That is AP2-G5 is also involved in regulating asexual development. However, on the other hand, this study did not provide enough data on the function of PbAP2-TR in the regulation of AP2-G, which was shown in the PfAP2-G5 study. The authors did not mention whether AP2-G was changed or not, whether gametocytogenesis and gametocyte development were changed. Based on the RNA-seq data (Table S5), AP2-G was slightly increased after PbAP2-TR was deleted at 12 hpi; meanwhile, AP2-G2 was significantly increased after PbAP2-TR KO (this may indicate that gametocyte development was changed).
2. If PbAP2-TR is involved in regulating AP2-G, the ChIP-seq and RNA-seq analysis after PbAP2-TR KO in AP2-G KO parasites may hide important information related to AP2-G. For example, PbAP2-TR binding sites in the AP2-G promoter (PfAP2-G5 was found to bind the AP2-G promoter) may be omitted.
3. The authors did not compare PbAP2-TR-regulated genes and binding motifs with the PfAP2-G5-regulated genes and binding motifs. At least, PbAP2-TR binding motif 2 is shared with PfAP2-G5 binding motif.
4. There is no "C" in fig. 4. The "D" to "G" should be "C" to "F".
5. Some critical data should be included in the manuscript. For example, the authors did not mention the number of biological replicates for RNA-seq.

Reviewer #4

(Remarks to the Author)

The authors clearly present an interesting and thorough study with largely compelling evidence that PbAP2-tr regulates asexual stage parasite genes by binding to two motifs at their promoters and by recruiting PbMORC to these sites where the complex acts to repress expression of these genes. This is an important and interesting finding. There are some issues with the way the data is presented, particularly the chip data, which hinders critical assessment and which could be altered to facilitate assessing the conclusions. The differential expression data should also be shown to not be affected by developmental delays which might have confounded the analyses.

Specific comments

For clarity I will use *-/-* to indicate a KOd allele throughout this review although I am aware that these parasites are haploid.

Page 5 line 166 “This result suggests that in *P. berghei*, PbAP2-TR is essential for asexual blood stage development and that repression of *ap2-g* transcription is not a common function of PbAP2-TR orthologs .”

This argument doesn't make sense to me, *ap2-tr* is KOd on a *ap2-g -/-* background and asexual *ap2-tr -/-* parasites are not recovered. This shows that AP2-tr is required for asexual growth as per the experiment performed on AP2-G *+/+* background. How do these experiments inform understanding of AP2-tr regulation of AP2-G? gametocyte production would need to be recorded to infer that AP2-tr was not regulating AP2-G. AP2-G is not essential so nothing can really be inferred from AP2-tr *-/-* AP2-g *-/-* parasite experiment.

Page 5 line 173 "To specifically assess the role of PbAP2-TR in asexual development, we disrupted pbap2-g in PbAP2-TR::GFP using the CRISPR/Cas9 system [PbAP2-TR::GFPpbap2-g(-), Fig S4]."

Although it does not affect interpretation of AP2-tr chip enrichment, it is unclear here why the chipseq was performed on ap2-g^{-/-} parasites. Later in the MS I realised that the authors were likely concerned that because PfAP2-G5 is an ortholog of AP2-tr and regulates gametocytogenesis so if AP2-tr also regulated gametocytogenesis then the resulting gametocytes could confound their analyses of asexual development. I think they should make this point explicitly before describing their use of AP2-G^{-/-} parasites because the rationale for using the AP2-G^{-/-} was unclear and seemed to relate more to disproving a role for AP2-tr in regulating gametocytogenesis, which was not done. It seems to me that the logical test would have been to compare gam counts between ap2-tr^{-/-} ap2-g^{-/-} and ap2-tr^{-/-}ap2-g^{+/+} to determine whether ap2-tr does actually regulate ap2-g. Is it not possible to count gams in Pb?

Figure 2A

Chipseq traces should include relevant input traces as well with II traces normalised fro library size. Alternatively chip/input could be shown and the raw traces for both reps chip and input could be provided as suppl figs.

Page 5 line 186

"Among the peaks containing both Motifs 1 and 2, the distance between these two motifs was not constant, suggesting that Motifs 1 and 2 function independently, rather than cooperatively, as a cis-regulatory element for PbAP2-TR (Fig 2F)."

A more cautious interpretation would be appropriate at this stage as it is not proven that ap2-tr binds either or both of the motifs, it is probable that other TFs bind these promoters, potentially within the same complex.

Page 6 lines 229-241 "PbAP2-TR functions as a transcriptional repressor"

A common confounder of inducible KO studies in plasmodium is developmental delay in the KO affecting maseq diff gene expression comparisons with the non-induced control. Have the matched ko and control maseq been assessed for developmental stage, e.g. using the mixture model of tonkin-hill et al 2018 plos biol.

Page 8 line 309 should this read fig 5b?

Fig 6C

Pbmorc and pbap2-tr chip traces should be shown as log₂chip/input with both chip and input somehow normlaised for library ize eg rpkm by a fixed bin size across the genome.

Fig 6D

Details should be provided of what is being shown, coverage is a generic term, it should be pbmorc log₂chip/input for rpkm by bin size or similar measure that accounts for library size and non specific enrichment ie input or non-specific precipitated material.

In general the chipseq data could be further explored, it would be interesting to see if Pb MORC and ap2-tr enrichment differ between heterochromatin and euchromatin and between coding sequence vs intergenic sequence using published datasets of heterochromatin marks eg H3K9me3 and input normalised data.

Version 1:

Reviewer comments:

Reviewer #1

(Remarks to the Author)

The authors have satisfactorily answered my queries.

Reviewer #2

(Remarks to the Author)

The authors have done fine and thorough work in revising their manuscript in response to the reviewers' comments. The study is clearly strengthened by the addition of new experimental data and by the substantial improvements in the analyses, particularly in the transcriptomics and ChIP-seq components. The textual revisions are appropriate and the authors have generally clarified their interpretations where needed.

Importantly, the new figures, controls, and re-analyses (including MA plots, QC metrics, reciprocal RIME, and improved visualization of AP2-TR/MORC occupancy) directly address the concerns raised during the first round of review. The expanded discussion and comparative analyses also improve the contextualization of the work within the broader ApiAP2/MORC literature.

Overall, I feel that the authors have adequately addressed all major and minor concerns. This is an interesting and well-executed study that provides the community with a solid mechanistic dissection of a trophozoite-stage transcriptional repressor and its cooperation with PbMORC during asexual blood-stage development. The manuscript now reaches a standard suitable for publication in its current form.

Reviewer #3

(Remarks to the Author)

The authors addressed all my concerns and comments. However, some responses were not satisfied and raised more questions.

In response to my #1 comment, authors confirmed that the expression of pbap2-G2 was increased and further indicated that PbAP2-TR ChIP-seq peaks were detected in the upstream region (at approximately 1.3 kb from the start codon) of pbap2-g2 with PbAP2-TR binding motifs around the summit. These data indicate that pbAP2-TR repress pbAP2-G2. Since pbAP2-G2 represses asexual proliferation and enhances gametocyte commitment and development, elevation of pbAP2-G2 expression after disrupt apAP2-TR could lead to downregulation of its target genes including many asexual genes (?)

In response to my #2 comment, authors indeed identified a pbAP2-TR ChIP-seq peak at about 2.5 upstream of pbAP2-G . This location is similar to the locations of PfAP-G and PfAP2-G5 ChIP-seq peaks at the upstream of PfAP2-G (Josling GA, Russell TJ, Venezia J, Orchard L, van Biljon R, Painter HJ, Llinás M. 2020. Dissecting the role of PfAP2-G in malaria gametocytogenesis. *Nat Commun* 11:1503; Shang X, Shen S, Tang J, He X, Zhao Y, Wang C, He X, Guo G, Liu M, Wang L, Zhu Q, Yang G, Jiang C, Zhang M, Yu X, Han J, Culleton R, Jiang L, Cao J, Gu L, Zhang Q. 2021. A cascade of transcriptional repression determines sexual commitment and development in *Plasmodium falciparum*. *Nucleic Acids Res* 49:9264–9279). More detailed analysis is required to check the peaks (such as motifs) between pb and pf. I still believe that conduction of the ChIP-seq and RNA-seq analysis after PbAP2-TR KO in the wildtype parasite but not in AP2-G KO parasites will provide more straightforward results.

Reviewer #4

(Remarks to the Author)

All of my comments have been addressed satisfactorily.

7/11/2025

Responses to Reviewers:

We would like to thank you for taking your time and effort to review our manuscript, “Precise gene regulation through transcriptional repression is essential for *Plasmodium* asexual blood stage development” (NCOMMS-25-48898). We found all of your comments highly constructive and insightful. Thus, we sincerely made revisions according to these comments.

Reviewer #1 (Remarks to the Author):

This manuscript describes the function of the Plasmodium berghei transcription factor the authors name Pb-TR.

The authors convincingly show that this AP2 domain-containing protein is essential for parasite development during the asexual replication cycle through a conditional KO. They go on to analyze its targets in the genome through both ChIP-Seq and DIP-Seq and convincingly identify candidate motifs. Interestingly, two independent motifs are identified and specify either the first two AP2 domains or the last single AP2 domain. The authors go on to determine that Pb-TR is more associated with genes identified in the ChIP-Seq analysis that are markers of early stages rather than late stages, leading to the hypothesis that Pb-TR is a repressor. They then further establish a phenotype using their conditional KO RNA-Seq, and show in a luciferase assay that the motifs identified can lead to higher reporter luciferase bioluminescence when mutated. Finally, they identify MORC as an interactor using both MS and by comparing ChIP-Seq profiles.

The data is overall compelling, and this is a useful addition to the understanding of transcription in the rodent model of malaria.

There are a some limitations of the study.

Thank you for your thorough review and constructive comments. Your advice led us to take a deeper look into our data and helped us improve the manuscript. We performed additional experiments and analyses and adjusted statements in the manuscripts according to comments made by all reviewers. We hope the changes made in the revised manuscript would satisfy your requirements below.

Given the strength of the phenotype, I found it surprising that the transcriptional changes in the RNA-Seq data following excision were somewhat limited and wondered

whether 12 hpi was indeed the right time point to conduct this assay. Only a minute number of genes with peaks are found to be upregulated. Is this low proportion expected, and is it sufficient to conclude that Pb-TR is a transcriptional repressor?

Thank you for your comment. About the gene upregulation in *pbap2-tr* KO parasites, the number of PbAP2-TR targets in the significantly upregulated genes (defined as $\log_2(\text{fold change}) > 1$ and $p\text{-value}^{\text{adj}} < 0.05$) were limited (47 genes) as you mention. However, 202 target genes were upregulated with $p\text{-value}^{\text{adj}} < 0.05$, and 73% of the targets had $\log_2(\text{fold change})$ higher than 0 (visualized as a histogram in Fig S6D, right). In addition, we could observe overall upregulation tendency for the target genes with a $p\text{-value}$ of 2.6×10^{-67} by two-tailed Student's t-test as described in the manuscript. We consider that this overall upregulation of target genes caused defects in the transcriptional control and eventually developmental arrest. These results are a strong indication that PbAP2-TR is a transcriptional repressor. Meanwhile, given your comment, we noticed that distribution of $\log_2(\text{fold change})$ values for target genes was not clear in the volcano plot in the main figure, as also mentioned by reviewer #2. Thus, we replaced the volcano plot in Fig 4A with the MA plot.

About the timing of differential expression analysis, the analysis at later time-point might detect more target genes in the significantly upregulated genes. However, development of *pbap2-tr* KO parasites is arrested at the trophozoite stage. After the developmental arrest, parasites would eventually go on towards cell death; thus, cell death-related biological processes (*e.g.* mRNA decay) might affect the transcriptomic analysis at the later time-point. Of note, reviewer #4 was also concerned about the possibility that developmental delay in the KO parasites has affected the differential expression analysis. Therefore, we chose 12 hpi, at which obvious developmental delay or arrest has not been observed in the *pbap2-tr* KO parasites (Fig 1E), to minimize such confounding effects in the analysis and to assess the transcriptomic changes directly induced by *pbap2-tr*-disruption.

In addition to the timing of performing the analysis, another factor is involved in the relatively low fold change values of some targets. As discussed in the manuscript, one of the major roles of PbAP2-TR is repression of genes that have peak expression during ring to early trophozoite development (*e.g.*, ribosome biogenesis and exported protein). These target genes are less responsive in the differential expression analysis because highly expressed genes show only modest fold changes when upregulated. Thus, we consider that the limited number of significantly upregulated target genes was due to their active transcription before PbAP2-TR expression.

Some of the analysis and the interpretations in the paper could be deeper. For instance, it would be interesting to dissect and discuss further the contribution of the different AP2 domains and whether they may be associated with different targets.

Thank you for your suggestion. Gene classification in the target genes that were associated with Motifs 1 or 2 was not largely different between them. In addition, no enrichment of Motif 1 or 2 targets was detected for any characteristic groups (p -value > 0.01 by Fisher's exact test for all characteristic groups). Given these, we added the following sentence and supplementary figure (Fig S7A) to describe these results and also included information of which motifs the targets are associated with in Table S3.

Line: "In any of these groups, no significant enrichment of targets associated with Motif 1 or 2 was detected, suggesting that difference in the binding motifs was not related to enrichment of characteristic groups in PbAP2-TR targets (Fig S7A and Table S3)."

Meanwhile, it is noteworthy that having different binding motifs (Motifs 1 and 2) is important for PbAP2-TR to generate variation in the level of transcriptional repression among its targets as we have shown in the differential expression analysis and luciferase reporter assays. Intriguingly, such a *cis* element-dependent mechanism of modulating the degree of transcriptional regulation was also found for PbSIP2 in our previous study. Thus, we have described these results in the Results section as follows.

Line: "These results verify that Motif 1 is a stronger *cis*-regulatory element for transcriptional repression by PbAP2-TR than Motif 2. Collectively, these results suggest that a single transcription factor can generate variations in the transcript levels of target genes through combinations of different *cis*-regulatory elements. In a previous study, we described a similar mechanism for transcriptional activation by PbSIP2²². These *cis*-element dependent variations in the levels of transcriptional activation and repression may be important for the parasite to regulate its complex life cycle with a small number of sequence-specific transcription factors¹⁹."

The discussion around the functional enrichment of Pb-TR targets is perhaps not deserving of a main figure, as it doesn't really enhance the understanding of the phenotype, and no attempt is made to use the RNA-Seq in tandem.

Thank you for your opinion. We think that it is very important to know what types of genes need to be repressed by PbAP2-TR for facilitating the trophozoite development. As we stated in the manuscript, through target classification, we unraveled that major

PbAP2-TR target groups were those that play a role in early blood stage development (e.g., ribosome biogenesis and exported protein). The time-course RNA-seq analysis further revealed their transcription patterns during ring-to-trophozoite development, which peaked at 8 hpi (Fig 5B and S7B). These results suggest that the major role of PbAP2-TR is to establish the transcriptional peak pattern during the ring to early trophozoite development. We consider that defects in controlling such transcriptional peak patterns of hundreds of its target genes caused the developmental arrest in the *pbap2-tr* KO parasite. Notably, the expression of these target-enriched gene groups was upregulated in the *pbap2-tr* KO parasites at 12 hpi. We added this result to the Results section with a supplementary figure Fig S7C as follows.

Line 332–333: “Notably, expression of targets in these groups was mostly upregulated in *pbap2-tr*-DiCre^{Rapa+} compared to *pbap2-tr*-DiCre^{Rapa-} (Fig S7C).”

Considering these results, we would like to present the characterization of PbAP2-TR targets in a main figure.

Another example is the other proteins identified by RIME, which include several chromatin-associated proteins, but no attempt is made at discussing whether they may also be present at Pb-TR sites and how that could play into the transcriptional regulation.

Thank you for your insight. As you mention, our RIME detected some proteins putatively related to transcriptional regulation in those uniquely IPed with PbAP2-TR::GFP, such as EELM2-domain containing protein and ISWI chromatin-remodeling complex ATPase. However, we were not sure to mention them as an interaction partner of PbAP2-TR because their average quantitative values (aveQVs) in PbAP2-TR::GFP ChIP samples were less than 3, significantly lower compared to that of PbAP2-TR (aveQV = 25.8) and PbMORC (aveQV = 21.7). Meanwhile, we agree that it is important to mention these proteins as potential interactors of PbAP2-TR (or PbMORC). In addition, we performed reciprocal RIME analysis using PbMORC::GFP^{*pbap2-g*(-)} (as suggested by reviewer #2) and detected several important transcription factors and putative regulators, in addition to above-mentioned factors. Collectively, we modified the Results and Discussion sections as follows. Also, we indicated these proteins in Fig 6A and 6C.

<Results>

Line 353–362: “*Plasmodium* parasites possess a single MORC, which has a conserved GHKL (gyrase, Hsp90, histidine kinase, and MutL)-ATPase domain and several Kelch motifs (Fig 6B)³⁴. To verify that PbMORC is a PbAP2-TR co-factor, we performed the

reciprocal RIME analysis at 12 hpi. To conduct this experiment on the *pbap2-g* knockout background as with the above RIME analysis, a parasite line expressing GFP-fused PbMORC was generated, and *pbap2-g* was disrupted in this parasite [PbMORC::GFP^{*pbap2-g*(-)}, Fig S8A]. The RIME analysis using PbMORC::GFP^{*pbap2-g*(-)} detected 146 proteins that were unique or more than five-fold enriched (Fig 6C and Table S8B). These proteins included PbMORC and PbAP2-TR with the first and third highest aveQVs of 121.3 and 47.5, respectively (Fig 6C and Table S8B), which supports the notion that PbMORC acts as a major co-factor of PbAP2-TR.”

Line 375–397: “Importantly, while ChIP-seq showed co-localization in the PbAP2-TR binding regions, only 38% (547/1436 peaks) of the PbMORC peaks overlapped with the PbAP2-TR peaks (Fig 6F). Motif enrichment analysis of the remaining PbMORC peaks (889 peaks) detected ACACA, TGCAAY, and YGTCT, along with their related motifs (Fig 6G and Table S9E). These results imply that PbMORC is recruited to the genome not only by PbAP2-TR but also through other factors. In the PbMORC RIME, two other AP2 family members, PBANKA_0521700 and PbAP2-P (PBANKA_0939100), were identified with the second and fourth highest aveQVs of 69.3 and 27.8, respectively (Fig 6C and Table S8B) while they were not detected in the PbAP2-TR RIME (Table S8A). These findings may suggest that PbMORC functions as a co-factor of PBANKA_0521700 and PbAP2-P during the IDC, independently of PbAP2-TR. Notably, in *P. falciparum*, the orthologs of these transcription factors, PF3D7_0420300 and PfAP2-P (PF3D7_1107800), were also detected in the co-IP with PfMORC^{34,35}. In addition, PF3D7_0420300 and PfAP2-P have been reported to bind to ACACA and TGCATG, respectively^{31,36}, which were analogous to those enriched in the PbMORC peaks.

RIME analyses of PbAP2-TR and PbMORC additionally identified several putative transcriptional regulators. The proteins commonly detected in the two RIME analyses included EELM2 (PBANKA_1234600) and ISWI (PBANKA_1123500) (Fig 6A, 6C, Table S8A, and S8B). Furthermore, other putative transcriptional regulators, such as CHD1 (PBANKA_0907200), ELM2 (PBANKA_0205000), and HDAC1 (PBANKA_0826500), were uniquely identified in the PbMORC RIME (Fig 6C and Table S8B). While their aveQVs in the RIME were relatively low compared to those of PbAP2-TR and PbMORC, these proteins might also contribute to the transcriptional regulation during the IDC as potential interaction partners of PbAP2-TR and/or PbMORC.”

<Discussion>

Line 450–464: “In *Toxoplasma*, MORC functions in a complex with histone deacetylase 3 (HDAC3) to regulate sexual commitment, suggesting that MORC functions are related to histone modifications⁵⁴. Comparably, HDAC1 was detected as an interaction partner

of both PbMORC and PfMORC^{34,35}, suggesting that the cooperation of MORC proteins with histone deacetylases is common in Apicomplexa. In contrast, RIME analysis with PbAP2-TR detected PbMORC as a major co-factor, but no histone-modifying enzymes. Similarly, our previous study revealed PbMORC as the only major co-factor of the female-specific transcriptional repressor complex, PbAP2-FG2/PbAP2R-2³⁹. These imply that, in *Plasmodium*, the components of co-factor complexes differ depending on the transcription factor that recruits them; *i.e.* when PbMORC is recruited by PbAP2-TR or PbAP2-FG2, they may repress target gene expression simply by remodeling chromatin states, without the involvement of histone modification. Alternatively, the PbAP2-TR and PbAP2-FG2 RIME did not detect histone modifying enzymes perhaps because they may initially act in a complex with MORC and subsequently dissociate from genomic DNA while MORC remains associated with chromatin. Following this dissociation, MORC may then recruit histone-modifying enzymes to further modulate the downstream gene expression.”

An interrogation arising from the paper is the contrast with the phenotype of the Plasmodium falciparum ortholog of Pb-TR. The authors contrast this in the discussion, but this could be really enhanced; a divergent role of an AP2 in different spp. is an interesting observation, and the authors could do more comparative analysis to establish whether there is a possible divergence. For instance, comparing targets of the APG5 and Pb-TR may be an interesting first step. Truly establishing a divergence between the two species would really enhance the impact of the paper.

Thank you for your suggestion. As you mention, it would be valuable to discuss a divergence in roles of PbAP2-TR and PfAP2-G5 through comparing their ChIP-seq results. In fact, we have previously downloaded the fastq files of PfAP2-G5 ChIP-seq data at the trophozoite stage and analyzed them using the same method we used in the current manuscript to compare them with our PbAP2-TR ChIP-seq data in detail. However, as we mapped their sequence data on the *P. falciparum* genome, their input data showed considerably high ratio of reads on gene bodies (input_1, 90.55%; input_2, 90.59%). Ideally, input data should show constant distribution of sequence reads across the genome; thus, the ratio of reads mapped on gene bodies should be comparable to the ratio of these regions on the genome (approximately 60% in *P. falciparum*). In addition, while ChIP data for transcription factors should have less sequence reads on gene bodies (because transcription factors bind to regulatory regions, which are mainly located on intergenic regions), more reads were mapped onto gene bodies for the PfAP2-G5 ChIP

compared with their input data (ChIP_1, 92.30%: ChIP_2, 92.40%). Furthermore, we noticed that read coverage of PfAP2-G5 ChIP and input data showed many peak-like patterns on gene bodies, unlike our ChIP data (Figure below). This sequence bias in the PfAP2-G5 ChIP-seq made us concerned about their data quality. Incidentally, our input data for PbAP2-TR showed 61% of reads on gene bodies, which was comparable to the ratio of gene body regions on the genome (62%). Furthermore, the ChIP data of PbAP2-TR showed smaller ratio of reads on gene bodies (46%).

The sequence bias in the PfAP2-G5 ChIP-seq seemed to be derived from GC bias (Figure below) and hence was probably caused by too many amplification cycles during the library preparation. (We suspect that amplification of extreme AT-rich sequences for the intergenic regions of *P. falciparum* is less efficient than gene body regions.) This bias must affect the peak calling results. In fact, more than half of peaks detected in our macs2 peak-calling for the PfAP2-G5 ChIP-seq were located on gene bodies, which was highly unusual for ChIP-seq of transcription factors. Presumably, due to such biased data, X Shang *et al.* defined genes with ChIP-seq peaks within their gene bodies as target genes, in addition to those with peaks upstream. This analysis is also unusual for ChIP-seq experiments of sequence-specific transcription factors.

Through the analysis, we considered that the ChIP-seq data of PfAP2-G5 was not performed with a good quality. This outcome made us hesitate to compare the ChIP-seq result with ours in detail and to discuss the roles of PbAP2-TR/PfAP2-G5 based on this comparison because we could not be sure if the results were derived from actual differences between the roles of PbAP2-TR and PfAP2-G5 or just sequence bias. However, as you asked (also asked by other reviewers), many readers would be interested in the comparison between the functions of PbAP2-TR and PfAP2-G5. Thus, we at least compared the target genes and binding motifs of PbAP2-TR with those of PfAP2-G5 reported in the study by X Shang *et al.* (for the target comparison, those with peaks upstream were compared). We described these as follows.

Line 223–225: “Among these targets, only 88 genes overlapped with those of PfAP2-G5 at the trophozoite stage (517 genes reported by Shang *et al.*) (Table S3).”

Line 194–195: “Similar motifs were also enriched in the ChIP-seq analysis of PfAP2-G5
30.”

Minor points:

A citation of a VEuPathDB publication should be included.

Thank you for your suggestion. We cited the following VEuPathDB publication at the sentence in which we mentioned PlasmoDB.

“Alvarez-Jarreta, J. et al. VEuPathDB: the eukaryotic pathogen, vector and host bioinformatics resource center in 2023. *Nucleic Acids Res* 52, D808–D816 (2024).”

Reviewer #2 (Remarks to the Author):

Summary: The study identifies PbAP2-TR (Trophozoite Repressor) as a trophozoite-stage AP2 transcription factor that functions as a transcriptional repressor essential for Plasmodium berghei asexual blood-stage development. Using DiCre conditional knockout, ChIP-seq/DIP-seq, time-course RNA-seq, motif mutagenesis luciferase reporters, and RIME/ChIP-seq for PbMORC, the authors show that PbAP2-TR binds two motifs (GTTGY and GTTCT) via distinct AP2 domains, represses target genes (notably ribosome biogenesis and exported proteins) from 8-16 hpi, and recruits PbMORC as a co-factor; only ~38% of MORC peaks overlap with PbAP2-TR, implying additional recruiters.

Thank you for your thorough review and constructive comments. Your advice on data analysis and visualization was very clear and helpful for us to improve our manuscript. Accordingly, we performed additional experiments and analyses and adjusted statements in the manuscripts. We hope the changes made in the revised manuscript would satisfy your requirements below.

Originality and field context: Prior work in Toxoplasma gondii established a MORC–HDAC3–ApiAP2 repression axis, and recent P. falciparum studies show PfMORC is essential, associates with heterochromatin, and co-IPs with multiple ApiAP2 proteins (including PfAP2-G5). The present study moves beyond association to a cis-encoded, stage-bounded mechanism in malaria parasites: a specific trophozoite ApiAP2 repressor (PbAP2-TR) with dual-motif/domain logic that recruits PbMORC to enforce a precise 8-16 hpi repression program. Notably, while PfAP2-G5 co-purifies with PfMORC, it is non-essential across the IDC and primarily linked to restraining gametocyte commitment, whereas PbAP2-TR is essential for asexual progression—underscoring a mechanistic advance rather than a mere extension of Pf co-IP observations. In short, the present story extends the Toxoplasma paradigm and fills a key mechanistic gap left by the PfMORC literature, making it a substantive, field-moving advance rather than an incremental confirmation.

Thank you for evaluating the originality of our manuscript in the field of apicomplexan biology. We are truly grateful with your generous and positive comments.

Major strengths

Clear stage specificity and genetic essentiality. PbAP2-TR is expressed in trophozoites and DiCre excision causes a trophozoite-stage developmental arrest with careful staging controls.

Cohesive multi-omics mechanism. ChIP-seq and DIP-seq define two motifs and assign them to different AP2 domains; luciferase mutagenesis shows cis-repressive function with Motif 1 > Motif 2 effect sizes.

Biological coherence. Target-enriched functional categories (ribosome biogenesis, exported proteins, pellicle/IMC) decline from 8-16 hpi, matching the proposed repressive window.

Thank you so much for captioning the major strengths in our manuscript. These statements show that you have reviewed our manuscript thoroughly. We really appreciate you taking your time to review our manuscript and provide these comments.

A) Key concerns about Transcriptomics (analysis & visualization)

The transcriptomic component is central to your mechanistic claim and is, overall, well integrated with the genetics and ChIP data. The authors analyze (i) an 8-16 hpi progression in an ap2-g- background to align with the trophozoite window of PbAP2-TR expression, and (ii) an acute pbap2-tr DiCre ± rapamycin contrast at 12 hpi with three biological replicates, mapped with HISAT2/featureCounts and modeled in DESeq2—good choices that match the biology and enable clear cross-validation between time course and inducible loss of function. At present, results are primarily visualized with volcano plots (Fig. 3A, Fig. 4A), and gene-set summaries are shown as TPM-scaled line plots across 4/8/16 hpi (Fig. 5B).

Use MA plots with shrunken log2FC values instead of volcano plots. The current volcano plots (Figs. 3A and 4A) overemphasize extreme p-values, which hides the role of expression strength and gives undue weight to low-count, highly significant genes. MA plots (mean expression vs shrunken log2FC, e.g. with DESeq2 apeglm/ashr) make effect sizes at biologically relevant expression levels clearer. Please re-render Figs. 3A and 4A as MA plots, marking FDR and |LFC| cutoffs directly; a simplified volcano can remain in the supplement if desired.

Thank you for your reasonable suggestion. Following your comment, we first applied DESeq2 apegglm to shrink the $\log_2(\text{fold change})$ values without filtering out genes with $\text{TPM} < 30$, which we did in the previous manuscript. We described this process in the Materials and methods section as follows and also cited A Zhu *et al.* (2018).

Line: “For differential expression analyses, DESeq2 apegglm shrinkage⁶⁵ was applied to the $\log_2(\text{fold change})$ to accurately evaluate the effect sizes without arbitrarily filtering out genes with low counts.”

Next, we produced MA plots for differential expression analyses of 8 vs 16 hpi and WT vs KO and replaced volcano plots in Fig 3A and 4A with these MA plots. For differential expression analysis between *pbap2-tr* KO and DMSO-control, we provided the volcano plot as a supplementary figure Fig S6D. For 8 vs 16 hpi, we did not include the volcano plot in a figure because many genes had p -values below 10^{-300} and were overemphasized in the plot as you mentioned.

Along with the apegglm application, results of differential expression analyses were slightly changed (*e.g.* the number of significantly up/downregulated genes, motif enrichment, and p -values for Student’s t-test). Accordingly, we modified some texts in the Results section and Fig 3B, 4B, and 4F.

Show full QC and replicate information. For the 12 hpi DiCre contrast (n = 3 per condition), include in the Supplement a PCA plot, a sample-to-sample distance heatmap, and the standard DESeq2 dispersion–mean plot to document replicate consistency and treatment separation. Clearly state the number of replicates for the 8/16 hpi time points and provide the same QC panels, so readers can be confident that the time-course results are robust.

Thank you for your advice. We provided the PCA plots, the sample-to-sample distance heatmaps, and the standard DESeq2 dispersion–mean plots for differential expression analyses of 8 vs 16 hpi and WT vs KO in the supplementary figures (Fig S5A–C and Fig S6A–C, respectively). Regarding these, we added the following sentence in the Results and Materials and methods sections.

Line 229–230: “(quality check results were shown in Fig S5A–C)”

Line 245–246: “(quality check results were shown in Fig S6A–C)”

Line 617–619: “To evaluate the replicate consistency, principal component analysis plots, sample-to-sample distance heatmaps, and dispersion plots were produced using DESeq2 plotPCA, dist, and plotDispEsts, respectively.”

In addition, we modified a sentence as follows to clarify the number of replicates

for the RNA-seq analysis of *pbap2-g* KO parasites at 8 and 16 hpi in the Results section. Line 227–229: “To obtain asexual transcriptomic data, RNA-seq analyses were conducted using gametocyte-less mutant parasites, *i.e.*, the *ap2-g*-knockout line, in biological triplicate”

Also, we added the following sentence to the figure caption of Fig 5B to clarify the number of replicates for the RNA-seq analysis of *pbap2-g* KO parasites at 4, 8, and 16 hpi.

Line 927–928: “Transcriptomic analyses were performed in biological triplicate at each time-point.”

Clarify the gene-set expression patterns. Fig. 5B’s line plots (TPM normalized to 4 hpi) are informative but overcrowded. Replace or supplement them with row-z-scored heatmaps of the main gene sets (“ribosome biogenesis,” “nucleolus,” “exported proteins/pellicle”) across 4/8/16 hpi, annotated with PbAP2-TR target status and Motif 1/Motif 2 presence near the TSS. This format will make the coordinated 8–16 hpi down-regulation immediately visible and directly link expression changes to cis-regulatory logic.

Thank you for your useful suggestion for improving our data presentation. Following your comment, we supplemented the Fig 5B with row-z-scored heatmaps, which were produced using Heatmapper (S Babicki et al., 2016) and were shown as a supplement figure Fig S7B. In the figure, we annotated PbAP2-TR targets on the side of heatmaps. About the motif presence, currently, TSSs in *P. berghei* have not been elucidated. Meanwhile, we think that regarding the *cis*-regulatory argument, whether these motifs exist in the PbAP2-TR binding regions would be a more appropriate criterion. Thus, we decided to categorize the targets according to whether Motifs 1 and 2 are present in the upstream ChIP-seq peak regions and to color-code them (those with both Motifs 1 and 2, only Motif 1, only Motif 2, and no motifs by yellow, green, blue, and grey, respectively).

B) Key concerns about MORC/AP2-TR

In Fig. 6C, the IGV snapshots don’t convincingly demonstrate co-localization: the two PbMORC replicates are shown on chromosome 1 with their own y-axis scales, and the PbAP2-TR track is plotted separately below, making direct, like-for-like visual comparison difficult; several MORC peaks appear without an obvious AP2-TR peak in the same view. This impression actually matches your own quantification that only 38% of MORC peaks overlap AP2-TR (Fig. 6E), so the panel, as drawn, tends to undercut

the text's claim of "genome-wide co-localization." Consider (i) replotting matched loci with identical y-axis scaling and a single, aligned multi-track view (MORC rep1/rep2 and AP2-TR together), (ii) adding zoom-ins on representative overlap and non-overlap sites, and (iii) emphasizing the more appropriate summary evidence (the heat map of MORC signal centered on AP2-TR peaks in Fig. 6D and the 38% overlap in Fig. 6E) while softening the wording from "genome-wide co-localization" to "co-localized at a subset of AP2-TR sites."

Thank you for your comment and ideas for improving our data presentation with detailed instructions. Considering these, we made the following modifications.

(i) We replaced the IGV images of row counts in ChIP-seq data into those of \log_2 -transformed ChIP/input and provided them as a single multi-track view (Fig 6D).

(ii) We added several IGV images at overlap and non-overlap sites as a supplementary figure Fig S8D (*rps24*, *rpl6*, and *imc1a* for overlap sites, *dlat*, *noc2*, and *cyt c* for non-overlap sites). In the figure, we also indicated positions of Motifs 1 and 2 by green and blue bars, respectively.

(iii) We indicated the 38%-overlap in Fig 6F. Regarding the co-localization of PbAP2-TR and PbMORC, we consider that these two factors indeed co-localize genome-wide. Although only 38% of PbMORC peaks overlapped with those of PbAP2-TR, these common peaks accounted for 54% of PbAP2-TR peaks. Given that even replicate ChIP-seq datasets typically show approximately 80% overlap under stringent peak-calling thresholds (*e.g.*, fold enrichment > 3 and *q*-value < 0.01 in our study), a 54% overlap between two distinct factors can be considered as substantially high. Meanwhile, the genome-wide co-localization cannot be fully demonstrated based on the peak-overlaps. Thus, we provided the heatmap of PbMORC ChIP-seq \log_2 (ChIP/input) across all PbAP2-TR peaks, which showed peak patterns around most PbAP2-TR peak summits (Fig 6E), including the regions defined as "non-overlap" sites (Fig S8D). This heatmap suggests that these "non-overlap" regions could not be detected as peaks in the PbMORC ChIP-seq due to our peak-calling criteria. In fact, when the PbMORC peaks were called with only the *q*-value threshold, they encompassed 90% of PbAP2-TR peaks. To make this clearer, we provided heatmap showing PbMORC \log_2 (ChIP/input) around the summit of "non-overlap" PbAP2-TR peaks (470 peaks, upper part of the Venn diagram in Fig S8C) as a supplementary figure Fig S8C. In addition, we modified the Results section as follows.

Line 366–374: "Among these common peaks, 547 peaks overlapped with those of PbAP2-TR, accounting for 54% (547/1017 peaks) of the PbAP2-TR peaks. Consistently,

Motifs 1 and 2 were significantly enriched in the PbMORC peak regions with p -values of 1.5×10^{-52} and 0.035, respectively, by Fisher's exact test (Table S9D). Furthermore, the PbMORC ChIP-seq coverage showed clear enrichment patterns around the summits of most PbAP2-TR peaks (Fig 6E), including those that were classified as "non-overlapping" with the PbMORC peaks under the thresholds of fold enrichment > 3 and q -value < 0.01 in macs2 peak calling (Fig S8C and S8D). These results indicate that PbAP2-TR genome-widely recruits PbMORC to the upstream regions of its target genes."

To move beyond correlative evidence of MORC co-recruitment, perform reciprocal co-IP or BioID to confirm direct complex formation and reinforce the central model.

Thank you for your suggestion. Accordingly, we performed RIME experiment using PbMORC::GFP^{pbap2-g(-)} and detected PbAP2-TR with the third highest average quantitative value (aveQV) (PbMORC being the highest), which strongly supports our conclusion that PbMORC is a major co-factor of PbAP2-TR. In addition, we detected two other AP2 transcription factors, PBANKA_0521700 and PbAP2-P, with the second and fourth highest aveQVs, respectively. Notably, in *P. falciparum*, these transcription factors were suggested to bind to the ACACA and TGCATG motifs, respectively, which were similar to those enriched in the PbMORC peak regions. Thus, the result suggests that these factors recruit PbMORC to specific *cis*-regulatory elements, other than PbAP2-TR binding motifs, on the genome.

In addition to these transcription factors, the PbMORC RIME detected several other transcriptional regulators (also described in the answer to your next comment). Of note, these included HDAC1, which was also detected previous co-IP with PfMORC in *P. falciparum*, indicating that the function of MORC is related to regulation of histone deacetylation and is conserved within Apicomplexa. Given this result and the RIME with PbAP2-TR, which did not detect HDAC1, we set up two hypotheses: 1. in *Plasmodium*, the components of co-factor complexes, which include MORC, are dependent on the transcription factor that recruits them, and for certain transcription factors, histone-modifying enzymes are not required. 2. transcription factors first recruit MORC and together regulate the downstream gene expression. After that, transcription factors dissociate from genomic DNA, and the remaining MORC subsequently recruits histone deacetylases to further modulate gene expression. Accordingly, we added Fig 6C and Table S8B for the PbMORC RIME results and modified the Results and Discussion sections as follows.

<Results>

Line 353–362: “*Plasmodium* parasites possess a single MORC, which has a conserved GHKL (gyrase, Hsp90, histidine kinase, and MutL)-ATPase domain and several Kelch motifs (Fig 6B)³⁴. To verify that PbMORC is a PbAP2-TR co-factor, we performed the reciprocal RIME analysis at 12 hpi. To conduct this experiment on the *pbap2-g* knockout background as with the above RIME analysis, a parasite line expressing GFP-fused PbMORC was generated, and *pbap2-g* was disrupted in this parasite [PbMORC::GFP^{*pbap2-g*(-)}, Fig S8A]. The RIME analysis using PbMORC::GFP^{*pbap2-g*(-)} detected 146 proteins that were unique or more than five-fold enriched (Fig 6C and Table S8B). These proteins included PbMORC and PbAP2-TR with the first and third highest aveQVs of 121.3 and 47.5, respectively (Fig 6C and Table S8B), which supports the notion that PbMORC acts as a major co-factor of PbAP2-TR.”

Line 375–397: “Importantly, while ChIP-seq showed co-localization in the PbAP2-TR binding regions, only 38% (547/1436 peaks) of the PbMORC peaks overlapped with the PbAP2-TR peaks (Fig 6F). Motif enrichment analysis of the remaining PbMORC peaks (889 peaks) detected ACACA, TGCAAY, and YGTCT, along with their related motifs (Fig 6G and Table S9E). These results imply that PbMORC is recruited to the genome not only by PbAP2-TR but also through other factors. In the PbMORC RIME, two other AP2 family members, PBANKA_0521700 and PbAP2-P (PBANKA_0939100), were identified with the second and fourth highest aveQVs of 69.3 and 27.8, respectively (Fig 6C and Table S8B) while they were not detected in the PbAP2-TR RIME (Table S8A). These findings may suggest that PbMORC functions as a co-factor of PBANKA_0521700 and PbAP2-P during the IDC, independently of PbAP2-TR. Notably, in *P. falciparum*, the orthologs of these transcription factors, PF3D7_0420300 and PfAP2-P (PF3D7_1107800), were also detected in the co-IP with PfMORC^{34,35}. In addition, PF3D7_0420300 and PfAP2-P have been reported to bind to ACACA and TGCATG, respectively^{31,36}, which were analogous to those enriched in the PbMORC peaks.

RIME analyses of PbAP2-TR and PbMORC additionally identified several putative transcriptional regulators. The proteins commonly detected in the two RIME analyses included EELM2 (PBANKA_1234600) and ISWI (PBANKA_1123500) (Fig 6A, 6C, Table S8A, and S8B). Furthermore, other putative transcriptional regulators, such as CHD1 (PBANKA_0907200), ELM2 (PBANKA_0205000), and HDAC1 (PBANKA_0826500), were uniquely identified in the PbMORC RIME (Fig 6C and Table S8B). While their aveQVs in the RIME were relatively low compared to those of PbAP2-TR and PbMORC, these proteins might also contribute to the transcriptional regulation during the IDC as potential interaction partners of PbAP2-TR and/or PbMORC.”

<Discussion>

Line 450–464: “In *Toxoplasma*, MORC functions in a complex with histone deacetylase 3 (HDAC3) to regulate sexual commitment, suggesting that MORC functions are related to histone modifications⁵⁴. Comparably, HDAC1 was detected as an interaction partner of both PbMORC and PfMORC^{34,35}, suggesting that the cooperation of MORC proteins with histone deacetylases is common in Apicomplexa. In contrast, RIME analysis with PbAP2-TR detected PbMORC as a major co-factor, but no histone-modifying enzymes. Similarly, our previous study revealed PbMORC as the only major co-factor of the female-specific transcriptional repressor complex, PbAP2-FG2/PbAP2R-2³⁹. These imply that, in *Plasmodium*, the components of co-factor complexes differ depending on the transcription factor that recruits them; *i.e.* when PbMORC is recruited by PbAP2-TR or PbAP2-FG2, they may repress target gene expression simply by remodeling chromatin states, without the involvement of histone modification. Alternatively, the PbAP2-TR and PbAP2-FG2 RIME did not detect histone modifying enzymes perhaps because they may initially act in a complex with MORC and subsequently dissociate from genomic DNA while MORC remains associated with chromatin. Following this dissociation, MORC may then recruit histone-modifying enzymes to further modulate the downstream gene expression.”

C) Minor comments

Your RIME list (Table S8) includes a protein bearing an ELM2 domain—precisely the class of ELM2–SANT co-repressors that scaffold MORC–HDAC3 complexes in Toxoplasma—yet the manuscript concludes that no chromatin regulators were detected and emphasizes MORC acting alone; please acknowledge this candidate and, ideally, test its association/function(reciprocal IP).

Thank you for your insight. As you mention, our RIME detected EELM2-domain containing protein in those uniquely IPed with PbAP2-TR::GFP. However, we were not sure to mention this result because its average quantitative value (aveQV) in PbAP2-TR::GFP ChIP samples were less than 3, significantly smaller compared to that of PbAP2-TR (aveQV = 25.8) and PbMORC (aveQV = 21.7). Meanwhile, we agree that it is important to discuss EELM2 as potential interactors of PbAP2-TR (or PbMORC) because as you mention, EELM2 (or ELM2) was previously identified in co-IP with MORC in *P. falciparum* and *Toxoplasma*. In addition, we performed reciprocal RIME analysis using PbMORC::GFP (as suggested above) and also detected EELM2. Accordingly, we modified the Results section as described above and indicated EELM2 in Fig 6A and 6C. In addition, we modified the Discussion section regarding MORC acting alone in

Plasmodium because, besides EELM2, we could detect HDAC1 in the PbMORC RIME as described in the answer to above comment.

ChIP-seq robustness: strengthen the ChIP-seq methods and reporting by providing replicate concordance and quality metrics (IDR, FRiP, NSC, RSC) for both AP2-TR and MORC datasets, clarifying how peaks were called and thresholds defined.

Thank you for your advice. According to your comment, we presented concordance in ChIP-seq peaks by performing IDR1D analysis (<https://idr2d.mit.edu/>) and showed the result in Fig S4C for PbAP2-TR and Fig S8B for PbMORC. In addition, to assess consistency in genome-wide ChIP/input values, we calculated Pearson correlation coefficient between the replicates as follows. We produced bigwig file of $\log_2(\text{ChIP}/\text{input})$ using bamCompare (deepTools) and then compared the replicate bigwig data at each 1 kb-bin using multiBigwigSummary bins (deepTools). These analyses yielded the Pearson correlation coefficient of 0.96 and 0.94 for PbAP2-TR and PbMORC ChIP-seq, respectively. The scatter plot for the comparison data was included in Fig 2A for PbAP2-TR and Fig 6D for PbMORC. Regarding these analyses, we modified the Results section and figure captions as follows.

<Results>

Line 184–186: “Genome-wide comparison of ChIP/input enrichment between the replicates resulted in Pearson correlation coefficient of 0.96, and IDR1D analysis revealed rank of peaks in each replicate to be largely consistent, showing high reproducibility of the data (Fig 2A, S4B, and S4C).”

Line 364–366: “of which 1436 peaks (77% of the experiment 1 peaks) overlapped (Pearson correlation coefficient = 0.94) (Fig 6D and S8B, and Table S9A, S9B, and S9C).”

<Figure captions>

Line 878–879: “A scatter plot at the right bottom shows comparison of $\log_2(\text{ChIP}/\text{input})$ in replicate at each 1-kb bin.”

Line 950–951: “A scatter plot on the right shows comparison of $\log_2(\text{ChIP}/\text{input})$ in replicate at each 1-kb bin.”

In addition, we calculated FRiP of all ChIP and input data using plotEnrichment (deepTools) and NSC and RSC using phantompeakqualtools. We summarized these values in Table S1C for PbAP2-TR and Table S9C for PbMORC. Accordingly, the following sentences were added in Materials and methods section.

Line 566–573: “The ChIP-seq enrichment on the genome was visualized as \log_2 -transformed ChIP/input ratio using bamCompare setting bin size as 10 bp. The replicate

data were compared using IDR1D (<https://idr2d.mit.edu/>)⁶¹, setting max gap as 100, and multiBigwigSummary bins, setting bin size as 1 kb, to assess their reproducibility. Fraction of reads in peaks (FRiP) was calculated using plotEnrichment, and normalized strand cross-correlation coefficient (NSC) and relative strand cross-correlation coefficient (RSC) were assessed using phantompeakqualtools for quality check of each ChIP and input data.”

About peak calling, we defined the parameters for macs2 in the Materials and methods section as follows.

Line 564–566: “peaks were called by the macs2 callpeak using the input sequence data as control (fold enrichment > 3.0, *q*-value < 0.01, and setting --call-summit option on).”

In addition, we added the following sentence to the Results section.

Line 181–182: “(by macs2 with the thresholds of fold enrichment > 3.0 and *q*-value < 0.01, using sequence data of input DNA as a control)”

Explicitly justify the use of the ±1200 bp promoter window in P. berghei, state whether spike-in normalization was applied, and include antibody validation details (lot, specificity) for the GFP ChIP. Finally, define the exact criteria for calling “targets” directly in the main text, not only in tables.

Thank you for very helpful point-by-point suggestions. About the 1,200-bp threshold for defining target genes, we have been using the criterion for several previous transcriptional regulator studies and found it useful. In the *P. berghei* genome, average size of intergenic regions is ca. 1600, and median is ca. 1400. In addition, in our previous studies of AP2-O and AP2-Sp (the master regulator of ookinete and sporozoite development, respectively), their binding motifs were observed within 1200 bp from the start codons of the major stage-specific genes. Based on these, we speculated that 1200 bp threshold for defining target genes would be sufficient to encompass most of the true targets. We think it is useful to include as many true targets as possible when evaluating the role of transcription factors based functional enrichment in targets. We added the following sentence to the Materials and methods section to describe these criteria in detail.

Line 582–584: “The 1200-bp threshold was determined based on the previous observations that binding motifs of AP2-O and AP2-Sp were detected within 1200 bp from the start codons of the major ookinete and sporozoite-specific genes, respectively^{62,63}”

We did not use spike-in normalization and thus added the following text to clarify. Line 550: “ChIP-seq was performed as previously described without using spike-in

normalization”

About anti-GFP antibody, we described the details in the reporting summary but forgot to include them in the main manuscript. Thank you for the notification. We added the details in the Materials and methods section as follows.

Line 555–556: “(Abcam, ab290, Rabbit polyclonal, reactive against all variants of *Aequorea victoria* GFP, GR19413-1)”

Criteria for target genes were described in the Results section as follows.

Line 222–223: “Based on ChIP-seq data, we identified 619 target genes of PbAP2-TR, which were defined as genes with a ChIP-seq peak within 1200 bp of the start codon (Table S3).”

D) Optional but valuable additions

Consider performing ATAC-seq in the PbAP2-TR KD context to directly link transcriptional repression with chromatin compaction at target loci.

Thank you for your suggestion. We agree that ATAC-seq analysis in *pbap2-tr* KO would be valuable because its co-factor PbMORC is a putative chromatin remodeling ATPase. Meanwhile, as our current manuscript and previous studies by others suggest, MORC is associated with several AP2 transcription factors during the *Plasmodium* asexual blood stage development. Hence, we are planning to investigate the chromatin-level regulation after evaluating the functions of other AP2 transcription factors that are associated with MORC, which we are currently pursuing, to capture the comprehensive regulatory mechanism during the asexual blood stage development. Thus, we would like to perform analyses on chromatin state, including ATAC-seq analysis, on WT and several AP2 gene KO parasites in the future.

Reviewer #3 (Remarks to the Author):

The authors described the function of PbAP2-TR, an AP2-domain containing transcriptional factor, in the regulation of malaria parasite asexual development. When this gene was conditionally knocked out, parasite growth was arrested at the trophozoite stage. ChIP-seq analysis revealed PbAP2-TR-regulated genes and their binding motifs. Differential expression analysis by RNA-seq indicated that PbAP2-TR is a transcriptional repressor, and this conclusion was confirmed by an episomal expression cassette with PbAP2-TR binding motifs. ChIP followed by mass spectrometry analysis revealed that PbMORC was associated with PbAP2-TR.

This work revealed the novel functions of PbAP2-TR, which is significant in the field.

Authors put a great effort to provide detailed information by advanced technologies such as Dicer-based conditional KO, ChIP-seq, and RNA-seq. The overall study is excellent and innovative. However, authors need to address quite a few questions to completely cover the whole story of this gene.

Thank you for your thorough review and constructive comments. Your insights into the concordance between PbAP2-TR and PfAP2-G5 were especially helpful for us to revise our data analysis. We performed additional experiments and analyses and adjusted statements in the manuscripts following comments from all reviewers. We hope the changes made in the revised manuscript would satisfy your requirements below.

1. First of all, this study revealed the potential hidden function of PfAP2-G5, the ortholog of PbAP2-TR in P. falciparum. That is AP2-G5 is also involved in regulating asexual development. However, on the other hand, this study did not provide enough data on the function of PbAP2-TR in the regulation of AP2-G, which was shown in the PfAP2-G5 study. The authors did not mention whether AP2-G was changed or not, whether gametocytogenesis and gametocyte development were changed. Based on the RNA-seq data (Table S5), AP2-G was slightly increased after PbAP2-TR was deleted at 12 hpi; meanwhile, AP2-G2 was significantly increased after PbAP2-TR KO (this may indicate that gametocyte development was changed).

Thank you for your important suggestion. Given your comment (and a comment from reviewer #4), we assessed the number of gametocytes in the *pbap2-tr*-DiCre^{Rapa-} and *pbap2-tr*-DiCre^{Rapa+} on Day 1 (24 hpi) (because gametocyte maturation takes approximately 24 h in *P. berghei*). The result showed no significant difference in gametocyte counts between *pbap2-tr*-DiCre^{Rapa-} and *pbap2-tr*-DiCre^{Rapa+} (Fig S3B). Thus, we described this result as follows.

Line 157–162: “we assessed the number of gametocytes in *pbap2-tr*-DiCre^{Rapa-} and *pbap2-tr*-DiCre^{Rapa+} on day 1 (24 hpi) of the above *in vivo* growth assays (because gametocyte maturation takes approximately 24 h in *P. berghei*) through Giemsa-staining. Gametocytemia (gametocyte/RBC) was comparable between *pbap2-tr*-DiCre^{Rapa-} and *pbap2-tr*-DiCre^{Rapa+}, demonstrating that the rate of commitment to gametocyte fate was not significantly affected by whether *pbap2-tr* was disrupted (Fig S3B).”

About the differential expression of *pbap2-g* in the *pbap2-tr* KO parasites, the log₂(fold change) and *p*-value for *pbap2-g* in the differential expression analysis were 0.075 and 0.79, respectively (after applying apeglm shrinkage as suggested by reviewer

#2). From these values, we cannot conclude that *pbap2-g* expression increased but rather think that we should state the *pbap2-g* expression has not been changed upon the *pbap2-tr*-disruption. Thus, we added the following sentence in the Results section.

Line 248–250: “Of note, *pbap2-g* was not differentially expressed in *pbap2-tr*-DiCre^{Rapa+} (log₂(fold change) = 0.075 and *p*-value^{adj} = 0.79) consistent with the comparable sexual commitment rate between *pbap2-tr*-DiCre^{Rapa-} and *pbap2-tr*-DiCre^{Rapa+} (Fig S3B).”

About the expression of *pbap2-g2*, its increase seemed to be directly induced by the *pbap2-tr*-disruption because the PbAP2-TR ChIP-seq peaks were detected in the upstream region (at approximately 1.3 kb from the start codon) of *pbap2-g2* with PbAP2-TR binding motifs around the summit (though *pbap2-g2* was not included in the targets due to the 1,200-bp threshold). We consider that the repression of *pbap2-g2* could be important for the asexual blood stage development, and thus, increase of its expression is not necessarily an indication of defects in gametocyte development.

2. If PbAP2-TR is involved in regulating AP2-G, the ChIP-seq and RNA-seq analysis after PbAP2-TR KO in AP2-G KO parasites may hide important information related to AP2-G. For example, PbAP2-TR binding sites in the AP2-G promoter (PfAP2-G5 was found to bind the AP2-G promoter) may be omitted.

Thank you for your comment. We understand your concern about overseeing the possible role of PbAP2-TR in regulating *pbap2-g*-transcription due to the experiments performed on the *pbap2-g* KO background. However, we performed the differential expression analysis in the WT background as described in the manuscript. In this analysis, we did not detect upregulation of *pbap2-g* in the *pbap2-tr* KO parasites (as described above), consistent with no increase of gametocyte production in the *pbap2-tr* KO parasites. This result suggests that PbAP2-TR does not play an important role in *pbap2-g* regulation.

About the ChIP-seq analysis in the *pbap2-g* KO background, if PbAP2-TR is important for the repression of *pbap2-g*, the ChIP-seq would still show a significant peak upstream *pbap2-g* because such repression should occur during the asexual blood stage development. In addition, we left the upstream region of *pbap2-g* intact in the *pbap2-g* KO parasites, so ChIP-seq peaks could be detected in this region. In fact, given your comment, we evaluated ChIP-seq peaks upstream *pbap2-g* and found a peak approximately 2.5 kb upstream from the start codon of *pbap2-g*, which was excluded in the target analysis due to the 1,200-bp threshold. Meanwhile, we detected no binding motifs near the peak summit. Together with that *pbap2-g* was not upregulated in the *pbap2-tr* KO parasites, we speculate that this PbAP2-TR binding is not important for its

function.

3. The authors did not compare PbAP2-TR-regulated genes and binding motifs with the PfAP2-G5-regulated genes and binding motifs. At least, PbAP2-TR binding motif 2 is shared with PfAP2-G5 binding motif.

Thank you for your comment. We agree with your suggestion that it would be valuable to discuss the roles of PbAP2-TR and PfAP2-G5 through comparing their target genes and binding motifs. In fact, we have previously downloaded the fastq files of PfAP2-G5 ChIP-seq data at the trophozoite stage and analyzed them using the same method we used in the current manuscript to compare them with our PbAP2-TR ChIP-seq data in detail. However, as we mapped their sequence data on the *P. falciparum* genome, their input data showed considerably high ratio of reads on gene bodies (input_1, 90.55%: input_2, 90.59%). Ideally, input data should show constant distribution of sequence reads across the genome; thus, the ratio of reads mapped on gene bodies should be comparable to the ratio of these regions on the genome (approximately 60% in *P. falciparum*). In addition, while ChIP data for transcription factors should have less sequence reads on gene bodies (because transcription factors bind to regulatory regions, which are mainly located on intergenic regions), more reads were mapped onto gene bodies for the PfAP2-G5 ChIP compared with their input data (ChIP_1, 92.30%: ChIP_2, 92.40%). Furthermore, we noticed that read coverage of PfAP2-G5 ChIP and input data showed many peak-like patterns on gene bodies, unlike our ChIP data (Figure below). This sequence bias in the PfAP2-G5 ChIP-seq made us concerned about their data quality. Incidentally, our input data for PbAP2-TR showed 61% of reads on gene bodies, which was comparable to the ratio of gene body regions on the genome (62%). Furthermore, the ChIP data of PbAP2-TR showed smaller ratio of reads on gene bodies (46%).

The sequence bias in the PfAP2-G5 ChIP-seq seemed to be derived from GC bias (Figure below) and hence was probably caused by too many amplification cycles during the library preparation. (We suspect that amplification of extreme AT-rich sequences for the intergenic regions of *P. falciparum* is less efficient than gene body regions.) This bias must affect the peak calling results. In fact, more than half of peaks detected in our macs2 peak-calling for the PfAP2-G5 ChIP-seq were located on gene bodies, which was highly unusual for ChIP-seq of transcription factors. Presumably, due to such biased data, X Shang *et al.* defined genes with ChIP-seq peaks within their gene bodies as target genes, in addition to those with peaks upstream. This analysis is also unusual for ChIP-seq experiments of sequence-specific transcription factors.

We next moved on to motif enrichment analysis for the PfAP2-G5 ChIP-seq. In the study by X Shang *et al.*, they only used peaks in the upstream regions of genes. In general, after defining peaks with certain criteria (*e.g.* high fold enrichment, *q*-values, and reproducibility), they should not be further filtered for motif enrichment analysis because unnecessarily peak filtering may result in missing some true binding motifs. Thus, we analyzed motif enrichment using all macs2-detected peaks (838 common peaks, fold enrichment > 3, *q*-value < 0.01). In the analysis, we detected enrichment of several GC-rich motifs (such as GCTGC and GGTGG), which were not detected in the study by X Shang *et al.* These motifs were more prevalent in gene bodies than intergenic regions, and when we assessed motif enrichment using only peaks on intergenic regions, these motifs were no longer enriched. Thus, we speculate that X Shang *et al.* decided to only use gene-upstream peaks because enrichment of these gene body-enriched motifs could be confounding.

Using the gene-upstream peaks, X Shang *et al.* detected five weblogo motifs, which included both Motifs 1 and 2-related motifs (GT[T/G]GTA and [T/G]GT[T/G]C, respectively). Among these five logos, two were highlighted in their manuscript and shown in the main figure. However, other three logos only appear in the supplementary figure, and the most enriched motif [A/G]CATGCA was not mentioned in the main manuscript or displayed in the main figure of the PfAP2-G5 study. Our motif enrichment analysis of PfAP2-G5 ChIP-seq detected several motifs related to four of these five weblogo motifs (ATATA was not detected). Among these motifs that are related to the four sequence logos, the most enriched motif was CATGCA (*p*-value = 3.5×10^{-104}) consistent with the analysis by X Shang *et al.* (Table below). Next, GTTGT and GTTGC (Motif 1) were enriched with *p*-values of 6.1×10^{-85} and 1.7×10^{-70} , respectively, and GTTCT (Motif 2) was detected with relatively moderate enrichment (*p*-value = 2.5×10^{-24}) (Table below). Enrichment of these motifs in both PbAP2-TR and PfAP2-G5 ChIP-seq analyses is reasonable because these motifs were also detected by our DIP-seq analyses, and amino acid sequences of AP2 domains are highly conserved among PbAP2-TR orthologs (96% and 87% for domains 1 and 2, respectively, and 100% for domain 3). These results suggest that PfAP2-G5 ChIP-seq peaks partially include regions that were actually bound by PfAP2-G5 through its AP2 domains.

Table. Motif analysis of PfAP2-G5 ChIP-seq

Motif by X Shang (DREME)	[A/G]CATGCA	GT[T/G]GTA (Motif 1-related)	[T/G]GT[T/G]C (Motif 2-related)	AA[A/C]AA
E-value	3.4×10^{-18}	1.2×10^{-3}	5.4×10^{-14}	2.7×10^{-14}

(DREME)				
Motif in our analysis (Fisher)	CATGCA	GTTGT and GTTGC (Motif 1)	GTTCT (Motif 2)	AACAA (TTGTT)
P -value (Fisher)	3.5×10^{-104}	6.1×10^{-85} and 1.7×10^{-70}	2.5×10^{-24}	1.9×10^{-8}

Another reported logo AACAA was enriched with a *p*-value of 1.9×10^{-8} in our analysis (Table above). However, the enrichment of AACAA (TTGTT) seemed to be derived from enrichment of Motifs 1 and 2 because it partially overlaps with both of these motifs. Moreover, in the analysis by X Shang *et al.*, AACAA was detected as AA[A/C]AA with A being slightly larger than C in the sequence logo. Thus, although X Shang *et al.* defined AACAA as the PfAP2-G5 binding motif and disrupted this motif upstream *pfap2-g*, it is very doubtful that AACAA actually functions as a *cis*-regulatory element recognized by PfAP2-G5 when it is not a part of Motifs 1 or 2.

Through the analysis, we considered that the ChIP-seq data of PfAP2-G5 was not performed with a good quality. This outcome made us hesitate to compare the ChIP-seq result with ours in detail and to discuss the roles of PbAP2-TR/PfAP2-G5 based on this comparison because we could not be sure if the differences in the result were derived from actual differences between the roles of PbAP2-TR and PfAP2-G5 or just sequence bias. However, as you asked (also asked by other reviewers), many readers would be interested in the comparison between the functions of PbAP2-TR and PfAP2-G5. Thus, we at least compared the target genes and binding motifs of PbAP2-TR with those of PfAP2-G5 reported in the study by X Shang *et al.* (for the target comparison, those with peaks upstream were compared). We described these as follows.

Line 223–225: “Among these targets, only 88 genes overlapped with those of PfAP2-G5 at the trophozoite stage (517 genes reported by Shang *et al.*) (Table S3).”

Line 194–195: “Similar motifs were also enriched in the ChIP-seq analysis of PfAP2-G5
30.”

4. There is no "C" in fig. 4. The "D" to "G" should be "C" to "F".

Thank you for kindly notifying the mislabel. We modified the figure caption accordingly.

5. Some critical data should be included in the manuscript. For example, the authors did not mention the number of biological replicates for RNA-seq.

Thank you for your comment. We described the number of replicates for RNA-seq

analysis in the Materials and methods section as “Three biologically independent experiments were performed”.

Meanwhile, we realized that the section only described the differential expression analysis between *pbap2-tr-DiCre*^{Rapa-} and *pbap2-tr-DiCre*^{Rapa+} but not the transcriptome analyses for *ap2-g* KO parasites at 4, 8, 16 hpi. Thus, we modified the Materials and methods section “RNA-seq and sequence data analysis” as follows.

Line 604–614: “For RNA-seq analysis using *pbap2-g*-knockout parasites, parasite cell cycles were synchronized as described in the parasite preparation. For RNA-seq analysis using *pbap2-tr-DiCre*, rapamycin-dependent disruption of *pbap2-tr* was induced, and schizonts of *pbap2-tr-DiCre*^{Rapa-} and *pbap2-tr-DiCre*^{Rapa+} were inoculated into mice, as described for the conditional knockout of *pbap2-tr*. Whole blood was harvested at 4, 8, and 16 hpi for *pbap2-g*-knockout parasites and at 12 hpi for *pbap2-tr-DiCre*^{Rapa-} and *pbap2-tr-DiCre*^{Rapa+} parasites. The blood was then passed through a Plasmodipur filter. RBCs were lysed in ice-cold 1.5 M NH₄Cl solution, and total RNA was extracted from the residual parasites using the Isogen II reagent (Nippon Gene). RNA-seq libraries were prepared using the KAPA mRNA HyperPrep Kit (Kapa Biosystems) and were sequenced using the MGI DNBSEQ-G400. Three biologically independent experiments were performed for each sample.”

Reviewer #4 (Remarks to the Author):

The authors clearly present an interesting and thorough study with largely compelling evidence that PbAP2-tr regulates asexual stage parasite genes by binding to two motifs at their promoters and by recruiting PbMORC to these sites where the complex acts to repress expression of these genes. This is an important and interesting finding. There are some issues with the way the data is presented, particularly the chip data, which hinders critical assessment and which could be altered to facilitate assessing the conclusions. The differential expression data should also be shown to not be affected by developmental delays which might have confounded the analyses.

Thank you for your thorough review and constructive comments. Your insights on ChIP-seq data presentation and potential issues in differential expression analysis were especially helpful for us to improve our manuscript. We performed additional experiments and analyses and modified statements in the manuscripts according to comments made by all reviewers. We hope the changes made in the revised manuscript would satisfy your requirements below.

Specific comments

For clarity I will use -/- to indicate a KO allele throughout this review although I am aware that these parasites are haploid.

Thank you for your indication.

Page 5 line 166 “This result suggests that in *P. berghei*, PbAP2-TR is essential for asexual blood stage development and that repression of *ap2-g* transcription is not a common function of PbAP2-TR orthologs .”

This argument doesn’t make sense to me, *ap2-tr* is KO on a *ap2-g* -/- background and asexual *ap2-tr* -/- parasites are not recovered. This shows that AP2-tr is required for asexual growth as per the experiment performed on AP2-G +/- background. How do these experiments inform understanding of AP2-tr regulation of AP2-G? gametocyte production would need to be recorded to infer that AP2-tr was not regulating AP2-G. AP2-G is not essential so nothing can really be inferred from AP2-tr-/- AP2-g-/- parasite experiment.

Thank you for your keen insight. In the PfAP2-G5 study, recovery of asexual growth rate in *pfap2-g5* KO parasites upon additional knockout of *pfap2-g* was an indication that PfAP2-G5 has a role in *pfap2-g* repression. Thus, to ensure that the loss of asexual proliferation in *pbap2-tr-DiCre^{Rapa-}* was not due to such effect and that PbAP2-TR is indeed essential for the asexual blood stage development, we performed the conditional knockout of *pbap2-tr* on the *pbap2-g* KO background. Meanwhile, as you mention, it was overinterpreting to conclude that PbAP2-TR does not play any role in *pbap2-g* regulation from this result. Thank you for pointing this out clearly. Given your following comments, we counted gametocytes in the *pbap2-tr* KO and the DMSO-control and observed no significant difference in the number of gametocytes on Day 1 (24 hpi). Considering your comment and this result, we modified the Result section as follows.

Line 157–172: “To evaluate whether PbAP2-TR plays a similar role, we assessed the number of gametocytes in *pbap2-tr-DiCre^{Rapa-}* and *pbap2-tr-DiCre^{Rapa+}* on day 1 (24 hpi) of the above *in vivo* growth assays (because gametocyte maturation takes approximately 24 h in *P. berghei*) through Giemsa-staining. Gametocytemia (gametocyte/RBC) was comparable between *pbap2-tr-DiCre^{Rapa-}* and *pbap2-tr-DiCre^{Rapa+}*, demonstrating that the rate of commitment to gametocyte fate was not significantly affected by whether *pbap2-tr* was disrupted (Fig S3B). Next, to further confirm that the loss of asexual proliferation in *pbap2-tr-DiCre^{Rapa+}* was not due to the parasite commitment to gametocyte fate, we

disrupted *pbap2-g* in *pbap2-tr-DiCre* using the CRISPR/Cas9 system [*pbap2-tr-DiCre^{pbap2-g(-)}*, Fig S3C]. DiCre-mediated disruption of *pbap2-tr* was induced in cultures (*pbap2-tr-DiCre^{pbap2-g(-)}_Rapa⁺* and *pbap2-tr-DiCre^{ap2-g(-)}_Rapa⁻*) (Genotyping 1, Fig S3D left), and after inoculating *pbap2-tr-DiCre^{pbap2-g(-)}_Rapa⁻* and *pbap2-tr-DiCre^{ap2-g(-)}_Rapa⁺* into mice, parasitemia was assessed daily. The result was similar to that of the phenotype analysis for *pbap2-tr-DiCre*; in *pbap2-tr-DiCre^{ap2-g(-)}_Rapa⁺*, parasitemia decreased from day 0 to day 1 and later increased (Fig S3E), but the day-3 parasites possessed the WT *pbap2-tr* locus (Genotyping 2, Fig S3D right). Collectively, these results suggest that PbAP2-TR is indeed essential for asexual blood stage development and does not play any evident role in the sexual commitment unlike PfAP2-G5.”

Page 5 line 173 “To specifically assess the role of PbAP2-TR in asexual development, we disrupted pbap2-g in PbAP2-TR::GFP using the CRISPR/Cas9 system [PbAP2-TR::GFPpbap2-g(-), Fig S4].”

Although it does not affect interpretation of AP2-tr chip enrichment, it is unclear here why the chipseq was performed on ap2-g-/- parasites. Later in the MS I realised that the authors were likely concerned that because PfAP2-G5 is an ortholog of AP2-tr and regulates gametocytogenesis so if AP2-tr also regulated gametocytogenesis then the resulting gametocytes could confound their analyses of asexual development. I think they should make this point explicitly before describing their use of AP2-G -/- parasites because the rationale for using the AP2-G -/- was unclear and seemed to relate more to disproving a role for AP2-tr in regulating gametocytogenesis, which was not done.

Thank you for your comment. In the PfAP2-G5 study, its role in early gametocyte development was also suggested besides regulating sexual commitment (repression of *pfap2-g*). While we could observe morphologically mature gametocytes in the *pbap2-tr* KO parasites, we were concerned that PbAP2-TR might play a role in early gametocyte development like PfAP2-G5 and that it might confound our analysis. Thanks to your comment, we realized that this was not clear in the manuscript, and hence, we modified the sentences as follows.

Line 177–178: “In *P. falciparum*, PfAP2-G5 also plays a role in the early gametocyte development. Thus, to specifically assess the role of PbAP2-TR in asexual development,”

It seems to me that the logical test would have been to compare gam counts between ap2-tr -/- ap2-g -/- and ap2-tr -/-ap2-g+/+ to determine whether ap2-tr does actually regulate ap2-g. Is it not possible to count gams in Pb?

Thank you for your kind suggestion. Given your comment, we assessed the number of gametocytes in the *pbap2-tr-DiCre*^{Rapa-} and *pbap2-tr-DiCre*^{Rapa+} on Day 1 (24 hpi) (because gametocyte maturation takes approximately 24 h in *P. berghei*). The result showed no significant difference in gametocyte counts between *pbap2-tr-DiCre*^{Rapa-} and *pbap2-tr-DiCre*^{Rapa+}. Consistent with this result, expression of *pbap2-g* was not upregulated in *pbap2-tr-DiCre*^{Rapa+} compared to *pbap2-tr-DiCre*^{Rapa-}. Thus, we think that we could at least state that the role of PbAP2-TR in *pbap2-g* repression is not as important as that of PfAP2-G5. Accordingly, we modified the Results section as described in the answer to your above comment.

Figure 2A

ChIPseq traces should include relevant input traces as well with ll traces normalised fro library size. Alternatively chip/input could be shown and the raw traces for both reps chip and input could be provided as suppl figs.

Thank you for your suggestion. We consider that showing raw coverage of ChIP data is useful to visualize the signal-to-noise ratio, which is important to evaluate the qualities of ChIP experiment and library preparation. This information would be lost when showing ChIP/input ratio although ChIP/input is useful to visualize the enrichment of ChIP signals from input data and to compare two different ChIP-seq experiments. In Fig 2A, we intend to show the ChIP quality of PbAP2-TR ChIP-seq data and reproducibility in duplicate. Thus, we would like to show ChIP data alone instead of ChIP/input in the main figure. Meanwhile, as you suggest, it is important to normalize the data by their library size and to present the coverage for the input data. Accordingly, we normalized the raw read coverage of ChIP and input data using bamCoverage (deepTools) and added four aligned traces of ChIP and input coverage data for duplicate as a supplementary figure Fig S4B. In addition, we modified the figure caption and Materials and methods as follows.

Line 876–877: “Histograms show the raw read coverage of ChIP data normalized by library size (bin size = 10 bp)”

Line 566–567: “The ChIP-seq enrichment on the genome was visualized as raw read coverage normalized by library size using bamCoverage (-bs 10, --normalizeUsing RPKM)”

Page 5 line 186

“Among the peaks containing both Motifs 1 and 2, the distance between these two motifs was not constant, suggesting that Motifs 1 and 2 function independently, rather than cooperatively, as a cis-regulatory element for PbAP2-TR (Fig 2F).”

A more cautious interpretation would be appropriate at this stage as it is not proven that ap2-tr binds either or both of the motifs, it is probable that other TFs bind these promoters, potentially within the same complex.

Thank you for your suggestion. Given your comment, we realize that this statement overestimates the relationship of Motif 1 and 2 in the context of PbAP2-TR functions at this stage. Hence, we modified the sentence as follows.

Line 199–200: *“i.e. these motifs are not arranged in a specific positional relationship (Fig 2F).”*

Page 6 lines 229-241 “PbAP2-TR functions as a transcriptional repressor”

A common confounder of inducible KO studies in plasmodium is developmental delay in the KO affecting rnaseq diff gene expression comparisons with the non-induced control. Have the matched ko and control rnaseq been assessed for developmental stage, e.g. using the mixture model of tonkin-hill et al 2018 plos biol.

Thank you for your kind caution. We were also afraid to detect differential gene expressions that occurred due to developmental delay or arrest caused by the conditional *pbap2-tr*-disruption. To minimize such confounding effects, we performed the differential expression analysis at 12 hpi, a time point well before any apparent developmental delay was observed in *pbap2-tr* KO parasites. (As described in the manuscript, ring-to-trophozoite ratio in the KO was comparable to that in the DMSO-control until 18 hpi (Fig 1E). Furthermore, the average size of trophozoites was also comparable between the KO and DMSO-control at 18 hpi (Fig 1F)). Therefore, we consider that differential gene expression detected in *pbap2-tr* KO at 12 hpi largely reflect the effect of *pbap2-tr*-disruption but not the effect of developmental delay.

Also, thank you for suggesting the use of the mixture model for estimating the population property in the RNA-seq samples. However, this statistic analysis performed by tonkin-hill *et al* requires template transcriptome data for each stage (*e.g.*, ring, trophozoite, schizont) to profile the parasite cell stage in the samples. In the public data, there are time-course transcriptome data during the asexual blood stage in *P. berghei*, but *pbap2-tr* KO trophozoite would not properly fit to the template trophozoite data because PbAP2-TR is a transcription factor, and disruption of its gene causes differential

expression of hundreds of genes, especially upregulation of its target genes. Thus, we think that use of the mixture model for our differential expression analysis would not be appropriate.

Page 8 line 309 should this read fig 5b?

Thank you for kindly notifying the mistake. Yes, it should be Fig 5b, and we modified it accordingly.

Fig 6C

Pbmorec and pbap2-tr chip traces should be shown as log2chip/input with both chip and input somehow normalised for library size eg rpkm by a fixed bin size across the genome.

Thank you for your suggestion. We replaced the ChIP-seq coverage to the log₂-transformed ChIP/input ratio and modified the figure caption and Materials and methods section as follows.

Line 948: “Histograms show the log₂-transformed ChIP/input ratio (bin size = 10 bp).

Line 566–568: “The ChIP-seq enrichment on the genome was visualized as raw read coverage normalized by library size using bamCoverage (-bs 10, --normalizeUsing RPKM) or log₂-transformed ChIP/input ratio using bamCompare (-bs 10).”

Fig 6D

Details should be provided of what is being shown, coverage is a generic term, it should be pbmorec log2chip/input for rpkm by bin size or similar measure that accounts for library size and non specific enrichment ie input or non-specific precipitated material.

Thank you for your comment. We are sorry for the ambiguity we made in the figure. Following your suggestion, we provided log₂(ChIP/input) for PbMORC ChIP-seq in the heatmap. In addition, we modified the figure caption as follows.

Line 951–952: “Heatmap showing log₂(ChIP/input) of PbMORC at the PbAP2-TR peaks.”

In general the chipseq data could be further explored, it would be interesting to see if Pb MORC and ap2-tr enrichment differ between heterochromatin and euchromatin and between coding sequence vs intergenic sequence using published datasets of heterochromatin marks eg H3K9me3 and input normalised data.

Thank you for your suggestion. Given your comment, we analyzed ChIP-seq data of PbHP1 (a marker for heterochromatin regions in *Plasmodium*) reported by K Witmer *et al.* (2020) and defined heterochromatin regions as 500-bp regions with $\text{RPM}^{\text{ChIP}}/\text{RPM}^{\text{input}} > 2$, obtaining 1294 regions. Within these regions, only 5 PbAP2-TR ChIP-seq peaks were detected. This number is too small, and thus, comparison of ChIP-seq enrichment within these regions did not seem to be very informative. For coding region vs intergenic region comparison, no significant difference was observed in the ChIP-seq enrichment plot of PbAP2-TR vs PbMORC (Figure below). This result was reasonable because PbAP2-TR and PbMORC genome-widely show similar peak patterns as depicted in the Fig 6E heatmap. For regions other than PbAP2-TR peaks, we think that the comparison of enrichment of these two factors would not be appropriate because while PbMORC also interacts with other transcription factors on the genome, PbAP2-TR is a sequence-specific transcription factor that directly binds to the genomic DNA.

16/12/2025

Responses to Reviewers:

We sincerely thank the reviewers for their careful evaluation of our revised manuscript, “Precise gene regulation through transcriptional repression is essential for *Plasmodium* asexual blood stage development” (NCOMMS-25-48898A). We are happy that most of the revisions satisfactorily addressed your previous concerns.

Reviewer #1 (Remarks to the Author):

The authors have satisfactorily answered my queries.

Thank you for your evaluation of our revised manuscript. We sincerely appreciate you taking your valuable time for the review process.

Reviewer #2 (Remarks to the Author):

The authors have done fine and thorough work in revising their manuscript in response to the reviewers' comments. The study is clearly strengthened by the addition of new experimental data and by the substantial improvements in the analyses, particularly in the transcriptomics and ChIP-seq components. The textual revisions are appropriate and the authors have generally clarified their interpretations where needed.

Importantly, the new figures, controls, and re-analyses (including MA plots, QC metrics, reciprocal RIME, and improved visualization of AP2-TR/MORC occupancy) directly address the concerns raised during the first round of review. The expanded discussion and comparative analyses also improve the contextualization of the work within the broader ApiAP2/MORC literature.

Overall, I feel that the authors have adequately addressed all major and minor concerns. This is an interesting and well-executed study that provides the community with a solid mechanistic dissection of a trophozoite-stage transcriptional repressor and its cooperation with PbMORC during asexual blood-stage development. The manuscript now reaches a standard suitable for publication in its current form.

Thank you for your evaluation of our previous revision. We sincerely appreciate you taking your valuable time to review our manuscript and to provide these comments. Your generous comments really encouraged us to further pursue our study.

Reviewer #3 (Remarks to the Author):

The authors addressed all my concerns and comments. However, some responses were not satisfied and raised more questions.

Thank you for your evaluation of our revised manuscript. We sincerely answered your following comments and modified the Discussion section.

In response to my #1 comment, authors confirmed that the expression of pbap2-G2 was increased and further indicated that PbAP2-TR ChIP-seq peaks were detected in the upstream region (at approximately 1.3 kb from the start codon) of pbap2-g2 with PbAP2-TR binding motifs around the summit. These data indicate that pbAP2-TR repress pbAP2-G2. Since pbAP2-G2 represses asexual proliferation and enhances gametocyte commitment and development, elevation of pbAP2-G2 expression after disrupt apAP2-TR could lead to downregulation of its target genes including many asexual genes (?)

Thank you for your comment. The primary role of PbAP2-G2 is to support sexual development after commitment to sexual fate (Yuda M, *et al.*, 2015). Accordingly, PbAP2-G2 targets encompass a wide variety of genes, including mosquito- and liver-stage-specific genes, not limited to genes promoting asexual replication in the blood. Given this broad spectrum of targets, it is reasonable to assume that the expression of PbAP2-G2 has a negative impact on development of the trophozoite, although it remains unclear to what extent PbAP2-G2 can function effectively during the trophozoite stage. Therefore, repression of *pbap2-g2* could be important for asexual development but would not account for the overall functions of PbAP2-TR. We consider that *pbap2-g2* as just one of the PbAP2-TR targets, rather than a dominant factor of the observed phenotypes.

In response to my #2 comment, authors indeed identified a pbAP2-TR ChIP-seq peak at about 2.5 upstream of pbAP2-G. This location is similar to the locations of PfAP-G and PfAP2-G5 ChIP-seq peaks at the upstream of PfAP2-G (Josling GA, Russell TJ, Venezia J, Orchard L, van Biljon R, Painter HJ, Llinás M. 2020. Dissecting the role of PfAP2-G in malaria gametocytogenesis. Nat Commun 11:1503; Shang X, Shen S, Tang J, He X, Zhao Y, Wang C, He X, Guo G, Liu M, Wang L, Zhu Q, Yang G, Jiang C, Zhang M, Yu X, Han J, Culleton R, Jiang L, Cao J, Gu L, Zhang Q. 2021. A cascade

of transcriptional repression determines sexual commitment and development in Plasmodium falciparum. Nucleic Acids Res 49:9264–9279). More detailed analysis is required to check the peaks (such as motifs) between pb and pf. I still believe that conduction of the ChIP-seq and RNA-seq analysis after PbAP2-TR KO in the wildtype parasite but not in AP2-G KO parasites will provide more straightforward results.

Thank you for your comment. As clarified in the previous revision, we performed the differential expression analysis between *pbap2-tr*-knockout parasites and DMSO-control on the WT background (not on the *pbap2-tr*-knockout background). In addition, we had provided rationale for why the *pbap2-g*-disruption would not affect the ChIP-seq results for evaluating the role of PbAP2-TR in regulation of *pbap2-g*, if any.

In general, ChIP-seq can detect non-specific signals in regions marked by constitutive heterochromatin. DNA fragments from these regions are poorly sheared by sonication and can become insoluble, making them prone to non-specific interaction with antibodies or beads. In ENCODE Blacklist for ChIP-seq in human and mouse, such regions are defined as regions that are recommended to be excluded from the analysis (Amemiya HM, Kundaje A, Boyle AP. The ENCODE Blacklist: Identification of Problematic Regions of the Genome. Sci Rep. 2019). Therefore, ChIP-seq peaks detected in these regions should be interpreted with caution, particularly when the peaks lack binding motifs. The upstream region of *pbap2-g* is marked with H3K9 trimethylation in asexual stage parasites. In addition, the PbAP2-TR ChIP-seq peaks upstream of *pbap2-g* did not contain PbAP2-TR binding motifs. Given these, we suspect that the PbAP2-TR ChIP peak in the upstream region of *pbap2-g* may be non-specific, and that PbAP2-TR does not play any role in gametocyte development. In fact, our differential expression analysis between *pbap2-tr*-knockout parasites and DMSO-control (on the WT background) showed that *pbap2-g* expression does not increase in *pbap2-tr*-knockout parasites. Furthermore, we did not detect any increase of gametocyte number in *pbap2-tr*-knockout parasites compared with the DMSO-control.

Reviewer #4 (Remarks to the Author):

All of my comments have been addressed satisfactorily.

Thank you for your evaluation of our revised manuscript. We sincerely appreciate you taking your valuable time for the review process.